# A State Representation for Diminishing Rewards

**Ted Moskovitz**
Gatsby Unit, UCL
`ted@gatsby.ucl.ac.uk`

**Samo Hromadka**
Gatsby Unit, UCL
`samo.hromadka.21@ucl.ac.uk`

**Ahmed Touati**
Meta
`atouati@meta.com`

**Diana Borsa**
Google DeepMind
`borsa@google.com`

**Maneesh Sahani**
Gatsby Unit, UCL
`maneesh@gatsby.ucl.ac.uk`

## Abstract

A common setting in multitask reinforcement learning (RL) demands that an agent rapidly adapt to various stationary reward functions randomly sampled from a fixed distribution. In such situations, the successor representation (SR) is a popular framework which supports rapid policy evaluation by decoupling a policy's expected discounted, cumulative state occupancies from a specific reward function. However, in the natural world, sequential tasks are rarely independent, and instead reflect shifting priorities based on the availability and subjective perception of rewarding stimuli. Reflecting this disjunction, in this paper we study the phenomenon of diminishing marginal utility and introduce a novel state representation, the $\lambda$ representation ($\lambda$R) which, surprisingly, is required for policy evaluation in this setting and which generalizes the SR as well as several other state representations from the literature. We establish the $\lambda$R's formal properties and examine its normative advantages in the context of machine learning, as well as its usefulness for studying natural behaviors, particularly foraging.

## 1   Introduction

The second ice cream cone rarely tastes as good as the first, and once all the most accessible brambles have been picked, the same investment of effort yields less fruit. In everyday life, the availability and our enjoyment of stimuli is sensitive to our past interactions with them. Thus, to evaluate different courses of action and act accordingly, we might expect our brains to form representations sensitive to the non-stationarity of rewards. Evidence in fields from behavioral economics [1, 2] to neuroscience [3] supports this hypothesis. Surprisingly, however, most of reinforcement learning (RL) takes place under the assumptions of the Markov Decision Process [MDP; 4], where rewards and optimal decision-making remain stationary.

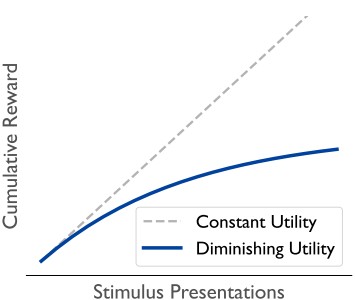

Figure 1.1: Diminishing rewards.

In this paper, we seek to bridge this gap by studying the phenomenon of *diminishing marginal utility* [5] in the context of RL. Diminishing marginal utility (DMU) is the subjective phenomenon by which repeated exposure to a rewarding stimulus reduces the perceived utility one experiences. While DMU is thought to have its roots in the maintenance of homeostatic equilibrium (too much ice cream can result in a stomach ache), it also manifests itself in domains in which the collected rewards are abstract, such as economics ($10 vs. $0 is perceived as a bigger difference in value than $1,010 vs. $1,000), where it is closely related to risk aversion [6, 7]. While DMU is well-studied in other fields, relatively few RL studies have explored diminishing reward functions [8, 9], and, to our

knowledge, none contain a formal analysis of DMU within RL. Here, we seek to characterize both its importance and the challenge it poses for current RL approaches (Section 3).

Surprisingly, we find that evaluating policies under diminishing rewards requires agents to learn a novel state representation which we term the $\lambda$ *representation* ($\lambda$R, Section 4). The $\lambda$R generalizes several state representations from the RL literature: the *successor representation* [SR; 10], the *first-occupancy representation* [FR; 11], and the *forward-backward representation* [FBR; 12], adapting them for non-stationary environments. Interestingly, despite the non-stationarity of the underlying reward functions, we show that the $\lambda$R still admits a Bellman recursion, allowing for efficient computation via dynamic programming (or approximate dynamic programming) and prove its convergence. We demonstrate the scalability of the $\lambda$R to large and continuous state spaces (Section 4.1), show that it supports policy evaluation, improvement, and composition (Section 5), and show that the behavior it induces is consistent with optimal foraging theory (Section 6).

## 2 Preliminaries

**Standard RL**   In standard RL, the goal of the agent is to act within its environment so as to maximize its discounted cumulative reward for some task $T$ [13]. Typically, $T$ is modeled as a discounted MDP, $T = (\mathcal{S}, \mathcal{A}, p, r, \gamma, \mu_0)$, where $\mathcal{S}$ is the state space, $\mathcal{A}$ is the set of available actions, $p : \mathcal{S} \times \mathcal{A} \mapsto \mathcal{P}(\mathcal{S})$ is the transition kernel (where $\mathcal{P}(\mathcal{S})$ is the space of probability distributions over $\mathcal{S}$), $r : \mathcal{S} \mapsto \mathbb{R}$ is the reward function, $\gamma \in [0, 1)$ is a discount factor, and $\mu_0 \in \mathcal{P}(\mathcal{S})$ is the distribution over initial states. Note that the reward function is also frequently defined over state-action pairs $(s, a)$ or triples $(s, a, s')$, but to simplify notation we mostly consider state-based rewards. All of the following analysis, unless otherwise noted, can easily be extended to the $(s, a)$- or $(s, a, s')$-dependent cases. The goal of the agent is to maximize its expected *return*, or discounted cumulative reward $\sum_t \gamma^t r(s_t)$. The role of the discount factor is twofold: from a theoretical standpoint, it ensures that this sum is finite for bounded rewards, and it induces myopia in the agent's behavior. To simplify notation, we will frequently write $r(s_t) \triangleq r_t$ and $\mathbf{r} \in \mathbb{R}^{|\mathcal{S}|}$ as the vector of rewards for each state. The agent acts according to a stationary policy $\pi : \mathcal{S} \mapsto \mathcal{P}(\mathcal{A})$. For finite MDPs, we can describe the expected transition probabilities under $\pi$ using a $|\mathcal{S}| \times |\mathcal{S}|$ matrix $P^\pi$ such that $P^\pi_{s,s'} = p^\pi(s'|s) \triangleq \sum_a p(s'|s, a)\pi(a|s)$. Given $\pi$ and a reward function $r$, the expected return associated with taking action $a$ in state $s$ is

$$Q_r^\pi(s, a) = \mathbb{E}_\pi \left[ \sum_{k=0}^\infty \gamma^k r_{t+k} \middle| s_t = s, a_t = a \right] = \mathbb{E}_{s' \sim p^\pi(\cdot|s)} \left[ r_t + \gamma Q_r^\pi(s', \pi(s')) \right]. \qquad (2.1)$$

The $Q_r^\pi$ are called the state-action values or simply the $Q$-values of $\pi$. The expectation $\mathbb{E}_\pi [\cdot]$ is taken with respect to the randomness of both the policy and the transition dynamics. For simplicity of notation, from here onwards we will write expectations of the form $\mathbb{E}_\pi [\cdot|s_t = s, a_t = a]$ as $\mathbb{E}_\pi [\cdot|s_t, a_t]$. The recursive form in Eq. (2.1) is called the *Bellman equation*, and it makes the process of estimating $Q_r^\pi$—termed *policy evaluation*—tractable via dynamic programming. In particular, successive applications of the *Bellman operator* $\mathcal{T}^\pi Q \triangleq r + \gamma P^\pi Q$ are guaranteed to converge to the true value function $Q^\pi$ for any initial real-valued $|\mathcal{S}| \times |\mathcal{A}|$ matrix $Q$. Once a policy has been evaluated, *policy improvement* identifies a new policy $\pi'$ such that $Q_r^\pi(s, a) \geq Q_r^{\pi'}(s, a), \ \forall(s, a) \in Q_r^\pi(s, a)$. Helpfully, such a policy can be defined as $\pi'(s) \in \operatorname{argmax}_a Q_r^\pi(s, a)$.

**Value Decomposition**   Often, it can be useful for an agent to evaluate the policies it's learned on new reward functions. In order to do so efficiently, it can make use of the *successor representation* [SR; 10], which decomposes the value function as follows:

$$V^\pi(s) = \mathbb{E}_\pi \left[ \sum_{k \geq 0} \gamma^k r_{t+k} \middle| s_t \right] = \mathbf{r}^\mathsf{T} \underbrace{\mathbb{E}_\pi \left[ \sum_{k \geq 0} \gamma^k \mathbf{1}(s_{t+k}) \middle| s_t \right]}_{\triangleq M^\pi(s)}, \qquad (2.2)$$

where $\mathbf{1}(s_t)$ is a one-hot representation of state $s_t$ and $M^\pi$, the SR, is an $|\mathcal{S}| \times |\mathcal{S}|$ matrix such that

$$M^\pi(s, s') \triangleq \mathbb{E}_\pi \left[ \sum_{k \geq 0} \gamma^k \mathbb{1}(s_{t+k} = s') \middle| s_t \right] = \mathbb{E}_\pi \left[ \mathbb{1}(s_t = s') + \gamma M^\pi(s_{t+1}, s') \middle| s_t \right]. \qquad (2.3)$$

Because the SR satisfies a Bellman recursion, it can be learned in a similar manner to the value function, with the added benefit that if the transition kernel $p$ is fixed, it can be used to instantly evaluate a policy on a task determined by any reward vector $\mathbf{r}$. The SR can also be conditioned on actions, $M^\pi(s, a, s')$, in which case multiplication by $\mathbf{r}$ produces the $Q$-values of $\pi$. The SR was originally motivated by the idea that a representation of state in the brain should be dependent on the similarity of future paths through the environment, and there is evidence that SR-like representations are present in the hippocampus [14]. A representation closely related to the SR is the *first-occupancy representation* [FR; 11], which modifies the SR by only counting the first visit to each state:

$$F^\pi(s, s') \triangleq \mathbb{E}_\pi \left[ \sum_{k \geq 0} \gamma^k \mathbb{1}(s_{t+k} = s', s' \notin \{s_t, \dots, s_{t+k-1}\}) \bigg| s_t \right]. \tag{2.4}$$

The FR can be used in the same way as the SR, with the difference that the values it computes are predicated on ephemeral rewards—those that are consumed or vanish after the first visit to a state.

**Policy Composition**  If the agent has access to a set of policies $\Pi = \{\pi\}$ and their associated SRs (or FRs) and is faced by a new task determined by reward vector $\mathbf{r}$, it can instantly evaluate these policies by simply multiplying: $\{Q^\pi(s, a) = \mathbf{r}^\mathsf{T} M^\pi(s, a, \cdot)\}$. This process is termed *generalized policy evaluation* [GPE; 15]. Similarly, *generalized policy improvement* (GPI) is defined as the identification of a new policy $\pi'$ such that $Q^{\pi'}(s, a) \geq \sup_{\pi \in \Pi} Q^\pi(s, a) \; \forall s, a \in \mathcal{S} \times \mathcal{A}$. Combining both provides a way for an agent to efficiently *compose* its policies $\Pi$—that is, to combine them to produce a new policy without additional learning. This process, which we refer to as GPE+GPI, produces the following policy, which is guaranteed to perform at least as well as any policy in $\Pi$ [16]:

$$\pi'(s) \in \operatorname*{argmax}_{a \in \mathcal{A}} \max_{\pi \in \Pi} \mathbf{r}^\mathsf{T} M^\pi(s, a, \cdot). \tag{2.5}$$

# 3  Diminishing Marginal Utility

**Problem Statement**  Motivated by DMU's importance in decision-making, our goal is to understand RL in the context of the following class of non-Markov reward functions:

$$r_\lambda(s, t) = \lambda(s)^{n(s,t)} \bar{r}(s), \quad \lambda(s) \in [0, 1], \tag{3.1}$$

where $n(s, t) \in \mathbb{N}$ is the agent's visit count at $s$ up to time $t$ and $\bar{r}(s)$ describes the reward at the first visit to $s$. $\lambda(s)$ therefore encodes the extent to which reward diminishes after each visit to $s$. Note that for $\lambda(s) = \lambda = 1$ we recover the usual stationary reward given by $\bar{r}$, and so this family of rewards strictly generalizes the stationary Markovian rewards typically used in RL.

**DMU is Challenging**  An immediate question when considering reward functions of this form is whether or not we can still define a Bellman equation over the resulting value function. If this is the case, standard RL approaches still apply. However, the following result shows otherwise.

**Lemma 3.1** (Impossibility; Informal). *Given a reward function of the form Eq. (3.1), it is impossible to define a Bellman equation solely using the resulting value function and immediate reward.*

We provide a more precise statement, along with proofs for all theoretical results, in Appendix B. This result means that we can't write the value function corresponding to rewards of the form Eq. (3.1) recursively only in terms of rewards and value in an analogous manner to Eq. (2.1). Nonetheless, we found that it *is* in fact possible to derive *a* recursive relationship in this setting, but only by positing a novel state representation that generalizes the SR and the FR, which we term the $\lambda$ *representation* ($\lambda$R). In the following sections, we define the $\lambda$R, establish its formal properties, and demonstrate its necessity for RL problems with diminishing rewards.

# 4  The $\lambda$ Representation

**A Representation for DMU**  We now the derive the $\lambda$R by decomposing the value function for rewards of the form Eq. (3.1) and show that it admits a Bellman recursion. To simplify notation, we

use a single $\lambda$ for all states, but the results below readily apply to non-uniform $\lambda$. We have

$$V^\pi(s) = \mathbb{E}\left[\sum_{k=0}^\infty \gamma^k r_\lambda(s_{t+k}, k)\big|s_t = s\right] = \bar{\mathbf{r}}^\mathsf{T} \underbrace{\mathbb{E}\left[\sum_{k=0}^\infty \gamma^k \lambda^{n_t(s_{t+k}, k)} \mathbf{1}(s_{t+k})\big|s_t = s\right]}_{\triangleq \Phi_\lambda^\pi(s)}, \qquad (4.1)$$

where we call $\Phi_\lambda^\pi(s)$ the $\lambda$ *representation* ($\lambda$R), and $n_t(s, k) \triangleq \sum_{j=0}^{k-1} \mathbb{1}(s_{t+j} = s)$, is the number of times state $s$ is visited from time $t$ up to—but not including—time $t + k$. Formally:

**Definition 4.1** ($\lambda$R). *For an MDP with finite $\mathcal{S}$ and $\boldsymbol{\lambda} \in [0, 1]^{|\mathcal{S}|}$, the $\lambda$ representation is given by $\Phi_\lambda^\pi$ such that*

$$\Phi_\lambda^\pi(s, s') \triangleq \mathbb{E}\left[\sum_{k=0}^\infty \lambda(s')^{n_t(s', k)} \gamma^k \mathbb{1}(s_{t+k} = s')\big|s_t = s\right] \qquad (4.2)$$

*where $n_t(s, k) \triangleq \sum_{j=0}^{k-1} \mathbb{1}(s_{t+j} = s)$ is the number of times state $s$ is visited from time $t$ until time $t + k - 1$.*

We can immediately see that for $\lambda = 0$, the $\lambda$R recovers the FR (we take $0^0 = 1$), and for $\lambda = 1$, it recovers the SR. For $\lambda \in (0, 1)$, the $\lambda$R interpolates between the two, with higher values of $\lambda$ reflecting greater persistence of reward in a given state or state-action pair and lower values of $\lambda$ reflecting more ephemeral rewards.

The $\lambda$R admits a recursive relationship:

$$\begin{aligned}
\Phi_\lambda^\pi(s, s') &= \mathbb{E}\left[\sum_{k=0}^\infty \lambda^{n_t(s', k)} \gamma^k \mathbb{1}(s_{t+k} = s')\Big|s_t = s\right] \\
&\overset{(i)}{=} \mathbb{E}\left[\mathbb{1}(s_t = s') + \lambda^{n_t(s', 1)}\gamma \sum_{k=1}^\infty \lambda^{n_{t+1}(s', k)} \gamma^{k-1} \mathbb{1}(s_{t+k} = s')\Big|s_t = s\right] \\
&= \mathbb{1}(s_t = s')(1 + \gamma\lambda\mathbb{E}_{s_{t+1}\sim p^\pi}\Phi_\lambda^\pi(s_{t+1}, s')) + \gamma(1 - \mathbb{1}(s_t = s'))\mathbb{E}_{s_{t+1}\sim p^\pi}\Phi_\lambda^\pi(s_{t+1}, s'),
\end{aligned} \qquad (4.3)$$

where $(i)$ follows from $n_t(s', k) = n_t(s', 1) + n_{t+1}(s', k-1)$. A more detailed derivation is provided in Appendix A. Thus, we can define a tractable Bellman operator:

**Definition 4.2** ($\lambda$R Operator). *Let $\Phi \in \mathbb{R}^{|\mathcal{S}|\times|\mathcal{S}|}$ be an arbitrary real-valued matrix, and let $\mathcal{G}_\lambda^\pi$ denote the $\lambda$R Bellman operator for $\pi$, such that*

$$\mathcal{G}_\lambda^\pi \Phi \triangleq I \odot \left(\mathbf{1}\mathbf{1}^\mathsf{T} + \gamma\lambda P^\pi\Phi\right) + \gamma(\mathbf{1}\mathbf{1}^\mathsf{T} - I) \odot P^\pi\Phi, \qquad (4.4)$$

*where $\odot$ denotes elementwise multiplication and $I$ is the $|\mathcal{S}| \times |\mathcal{S}|$ identity matrix. In particular, for a stationary policy $\pi$, $\mathcal{G}_\lambda^\pi \Phi_\lambda^\pi = \Phi_\lambda^\pi$.*

The following result establishes that successive applications of $\mathcal{G}_\lambda^\pi$ converge to the $\lambda$R.

**Proposition 4.1** (Convergence). *Under the conditions assumed above, set $\Phi^{(0)} = (1 - \lambda)I$. For $k = 1, 2, \ldots$, suppose that $\Phi^{(k+1)} = \mathcal{G}_\lambda^\pi \Phi^{(k)}$. Then*

$$|(\Phi^{(k)} - \Phi_\lambda^\pi)_{s,s'}| \leq \frac{\gamma^{k+1}}{1 - \lambda\gamma}.$$

While such analysis is fairly standard in MDP theory, it is noteworthy that the analysis extends to this case despite Lemma 3.1. That is, for a ethologically relevant class of *non-Markovian* reward functions [17], it is possible to define a Markovian Bellman operator and prove that repeated applications of it converge to the desired representation. Furthermore, unlike in the stationary reward case, where decomposing the value function in terms of the reward and the SR is "optional" to perform prediction or control, in this setting the structure of the problem *requires* that this representation be learned. To get an intuition for the $\lambda$R consider the simple gridworld presented in Fig. 4.1, where we visualize the $\lambda$R for varying values of $\lambda$. For $\lambda = 0$, the representation recovers the FR, encoding the expected discount at first occupancy of each state, while as $\lambda$ increases, effective occupancy is accumulated accordingly at states which are revisited. We offer a more detailed visualization in Fig. F.2.

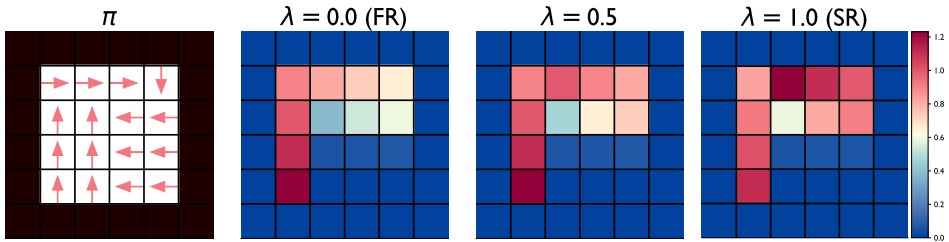

Figure 4.1: **The $\lambda$R interpolates between the FR and the SR.** We visualize the $\lambda$Rs of the bottom left state for the depicted policy for $\lambda \in \{0.0, 0.5, 1.0\}$.

## 4.1 Continuous State Spaces

When the state space is large or continuous, it becomes impossible to use a tabular representation, and we must turn to function approximation. There are several ways to approach this, each with their own advantages and drawbacks.

**Feature-Based Representations**   For the SR and the FR, compatibility with function approximation is most commonly achieved by simply replacing the indicator functions in their definitions with a *base feature function $\phi : \mathcal{S} \mapsto \mathbb{R}^D$* to create *successor features* [SFs; 15, 16] and *first-occupancy features* [FFs; 11], respectively. The intuition in this case is that $\phi$ for the SR and the FR is just a one-hot encoding of the state (or state-action pair), and so for cases when $|\mathcal{S}|$ is too large, we can replace it with a compressed representation. That is, we have the following definition

**Definition 4.3** (SFs). *Let $\phi : \mathcal{S} \mapsto \mathbb{R}^D$ be a base feature function. Then, the successor feature (SF) representation $\varphi_1 : \mathcal{S} \times \mathcal{A} \mapsto \mathbb{R}^D$ is defined as $\varphi_1^\pi(s,a) \triangleq \mathbb{E}_\pi \left[ \sum_{k=0}^\infty \gamma^k \phi(s_{t+k}) \middle| s_t, a_t \right]$ for all $s, a \in \mathcal{S} \times \mathcal{A}$.*

One key fact to note, however, is that due to this compression, all notion of state "occupancy" is lost and these representations instead measure feature accumulation. For any feasible $\lambda$ then, it is most natural to define these representations using their recursive forms:

**Definition 4.4** ($\lambda$F). *For $\lambda \in [0, 1]$ and bounded base features $\phi : \mathcal{S} \mapsto [0, 1]^D$, the $\lambda$-feature ($\lambda$F) representation of state $s$ is given by $\varphi_\lambda^\pi$ such that*

$$\varphi_\lambda^\pi(s) \triangleq \phi(s) \odot \left(1 + \gamma\lambda\mathbb{E}_{s'\sim p^\pi(\cdot|s)}\varphi_\lambda^\pi(s')\right) + \gamma(1 - \phi(s)) \odot \mathbb{E}_{s'\sim p^\pi(\cdot|s)}\varphi_\lambda^\pi(s'). \quad (4.5)$$

In order to maintain their usefulness for policy evaluation, the main requirement of the base features is that the reward should lie in their span. That is, a given feature function $\phi$ is most useful for an associated set of reward functions $\mathcal{R}$, given by

$$\mathcal{R} = \{r \mid \exists \mathbf{w} \in \mathbb{R}^D \text{ s.t. } r(s,a) = \mathbf{w}^\mathsf{T}\boldsymbol{\phi}(s,a) \; \forall s, a \in \mathcal{S} \times \mathcal{A}\}. \quad (4.6)$$

However, Barreto et al. [18] demonstrate that good performance can still be achieved for an arbitrary reward function as long as it's sufficiently close to some $r \in \mathcal{R}$.

**Set-Theoretic Formulations**   As noted above, computing expectations of accumulated abstract features is unsatisfying because it requires that we lose the occupancy-based interpretation of these representations. It also restricts the agent to reward functions which lie within $\mathcal{R}$. An alternative approach to extending the SR to continuous MDPs that avoids this issue is the *successor measure* [SM; 19], which treats the distribution of future states as a measure over $\mathcal{S}$:

$$M^\pi(s, a, X) \triangleq \sum_{k=0}^\infty \gamma^k \mathbb{P}(s_{t+k} \in X \mid s_t = s, a_t = a, \pi) \quad \forall X \subset \mathcal{S} \text{ measurable}, \quad (4.7)$$

which can be expressed in the discrete case as $M^\pi = I + \gamma P^\pi M^\pi$. In the continuous case, matrices are replaced by their corresponding measures. Note that SFs can be recovered by integrating: $\varphi_1^\pi(s, a) = \int_{s'} M^\pi(s, a, ds')\phi(s')$. We can define an analogous object for the $\lambda$R as follows

$$\Phi_\lambda^\pi(s, X) \triangleq \sum_{k=0}^\infty \lambda^{n_t(X,k)}\gamma^k \mathbb{P}(s_{t+k} \in X \mid s_t = s, \pi) \quad (4.8)$$

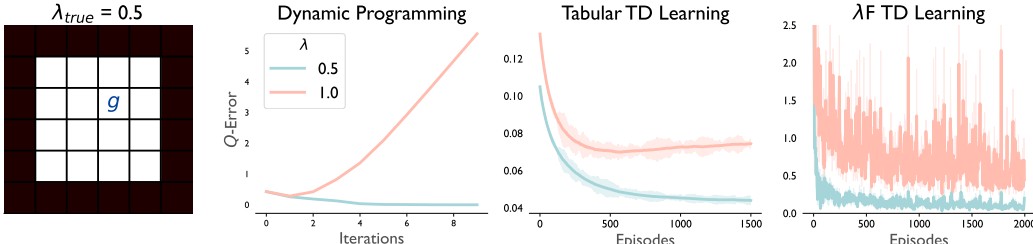

Figure 5.1: **The $\lambda$R is required for accurate policy evaluation.** Policy evaluation of the policy depicted in Fig. 4.1 using dynamic programming, tabular TD learning, and $\lambda$F TD learning produces the most accurate value estimates when using the $\lambda$R with $\lambda = \lambda_{true}$. Results are averaged over three random seeds. Shading indicates standard error.

where $n_t(X, k) \triangleq \sum_{j=0}^{k-1} \delta_{s_{t+j}}(X)$. However, this is not a measure because it fails to satisfy additivity for $\lambda < 1$, i.e., for measurable disjoint sets $A, B \subseteq \mathcal{S}$, $\Phi_\lambda^\pi(s, A \cup B) < \Phi_\lambda^\pi(s, A) + \Phi_\lambda^\pi(s, B)$ (Lemma B.4). For this reason, we call Eq. (4.8) the $\lambda$ *operator* ($\lambda$O). We then minimize the following squared Bellman error loss for $\Phi_\lambda^\pi$ (dropping sub/superscripts for concision):

$$\mathcal{L}(\Phi) = \mathbb{E}_{s_t, s_{t+1} \sim \rho, s' \sim \mu} \left[ (\varphi(s_t, s') - \gamma \bar{\varphi}(s_{t+1}, s'))^2 \right] - 2 \mathbb{E}_{s_t \sim \rho} [\varphi(s_t, s_t)] \\ + 2\gamma(1-\lambda) \mathbb{E}_{s_t, s_{t+1} \sim \rho} [\mu(s_t) \varphi(s_t, s_t) \bar{\varphi}(s_{t+1}, s_t)], \tag{4.9}$$

where $\rho$ and $\mu$ are densities over $\mathcal{S}$ and $\Phi_\lambda^\pi(s) \triangleq \varphi_\lambda^\pi(s) \mathrm{diag}(\mu)$ in the discrete case, with $\varphi_\lambda^\pi$ parametrized by neural network. $\bar{\cdot}$ indicates a `stop-gradient` operation, i.e., a target network. A detailed derivation and discussion are given in Appendix H. While $\rho$ can be any training distribution of transitions we can sample from, we require an analytic expression for $\mu$. Eq. (4.9) recovers the SM loss of Touati and Ollivier [12] when $\lambda = 1$.

## 5 Policy Evaluation, Learning, and Composition under DMU

In the following sections, we experimentally validate the formal properties of the $\lambda$R and explore its usefulness for solving RL problems with DMU. The majority of our experiments center on navigation tasks, as we believe this is the most natural setting for studying behavior under diminishing rewards. However, in Appendix I we also explore potential for the $\lambda$R's use other areas, such as continuous control, even when rewards do not diminish. There is also the inherent question of whether the agent has access to $\lambda$. In a naturalistic context, $\lambda$ can be seen as an internal variable that the agent likely knows, especially if the agent has experienced the related stimulus before. Therefore, in subsequent experiments, treating $\lambda$ as a "given" can be taken to imply the agent has prior experience with the relevant stimulus. Further details for all experiments can be found in Appendix E.

### 5.1 Policy Evaluation

In Section 4, we showed that in order to perform policy evaluation under DMU, an agent is required to learn the $\lambda$R. In our first experimental setting, we verify this analysis empirically for the policy depicted in Fig. 4.1 with a rewarded location in the state indicated by a $g$ in Fig. 5.1 with $\lambda_{true} = 0.5$. We then compare the performance for agents using different values of $\lambda$ across dynamic programming (DP), tabular TD learning, and $\lambda$F TD learning with Laplacian features. For the latter two, we use a linear function approximator with a one-hot encoding of the state as the base feature function. We then compute the $Q$-values using the $\lambda$R with $\lambda \in \{0.5, 1.0\}$ (with $\lambda = 1$ corresponding to the SR) and compare the resulting value estimates to the true $Q$-values. Consistent with our theoretical analysis, Fig. 5.1 shows that the $\lambda$R with $\lambda = \lambda_{true}$ is required to produce accurate value estimates.

### 5.2 Policy Learning

To demonstrate that $\lambda$R is useful in supporting policy improvement under diminishing rewards, we implemented modified forms of $Q$-learning [20] (which we term $Q_\lambda$-learning) and advantage

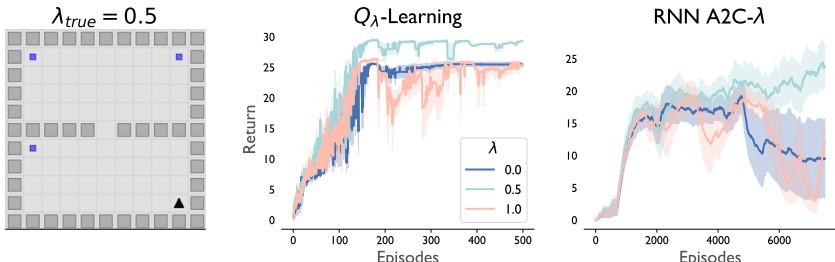

Figure 5.2: **The $\lambda$R is required for strong performance.** We apply a tabular $Q$-learning-style algorithm and deep actor-critic algorithm to policy optimization in the TwoRooms domain. The blue locations indicate reward, and the black triangle shows the agent's position. Results are averaged over three random seeds. Shading indicates standard error.

actor-critic [A2C; 21] and applied them to the TwoRooms domain from the NeuroNav benchmark task set [22] (see Fig. 5.2). For a transition $(s_t, a_t, s_{t+1})$, we define the following operator:

$$\mathcal{T}_\lambda^\star \Phi \triangleq \mathbf{1}(s_t) \odot (1 + \lambda\gamma\Phi(s_{t+1}, a_{t+1})) + \gamma(1 - \mathbf{1}(s_t)) \odot \Phi(s_{t+1}, a_{t+1}), \tag{5.1}$$

where $a_{t+1} = \operatorname{argmax}_a Q_\lambda(s_t, a) = \operatorname{argmax}_a \mathbf{r}^\mathsf{T}\Phi(s_t, a)$. This is an improvement operator for $Q_\lambda$. The results in Fig. 5.2 show that $Q_\lambda$-learning outperforms standard $Q$-learning ($\lambda = 1$) for diminishing rewards, and that the "correct" $\lambda$ produces the best performance. To implement A2C with a $\lambda$R critic, we modified the standard TD target in a similar manner as follows:

$$\mathcal{T}_\lambda V(s_t) = r(s_t) + \gamma(V(s_{t+1}) + (\lambda - 1)\mathbf{w}^\mathsf{T}(\phi(s_t) \odot \varphi_\lambda(s_{t+1}))), \tag{5.2}$$

where $\phi$ were one-hot state encodings, and the policy, value function, and $\lambda$F were output heads of a shared LSTM network [23]. Note this target is equivalent to Definition 4.4 multiplied by the reward (derivation in Appendix E). Fig. 5.2 shows again that correct value targets lead to improved performance. Videos of agent behavior can be found at `lambdarepresentation.github.io`.

## 5.3 Policy Composition

As we've seen, DMU problems of this form have an interesting property wherein solving one task requires the computation of a representation which on its own is task agnostic. In the same way that the SR and FR facilitate generalization across reward functions, the $\lambda$R facilitates generalization across reward functions with different $\bar{r}$s. The following result shows that there is a benefit to having the "correct" $\lambda$ for a given resource.

**Theorem 5.1** (GPI). *Let $\{M_j\}_{j=1}^n \subseteq \mathcal{M}$ and $M \in \mathcal{M}$ be a set of tasks in an environment $\mathcal{M}$ and let $Q^{\pi_j^*}$ denote the action-value function of an optimal policy of $M_j$ when executed in $M$. Assume that the agent uses diminishing rate $\hat{\lambda}$ that may differ from the true environment diminishing rate $\lambda$. Given estimates $\tilde{Q}^{\pi_j}$ such that $\|Q^{\pi_j^*} - \tilde{Q}^{\pi_j}\|_\infty \leq \epsilon$ for all $j$, define*

$$\pi(s) \in \operatorname*{argmax}_a \max_j \tilde{Q}^{\pi_j}(s, a).$$

*Then,*

$$Q^\pi(s, a) \geq \max_j Q^{\pi_j^*}(s, a) - \frac{1}{1 - \gamma}\left(2\epsilon + |\lambda - \hat{\lambda}|\|r\|_\infty + \frac{\gamma(1 - \lambda)r(s, a)}{1 - \lambda\gamma}\right).$$

Note that for $\lambda = 1$, we recover the original GPI bound due to Barreto et al. [18] with an additional term quantifying error accrued if incorrectly assuming $\lambda < 1$.

**Tabular Navigation** We can see this result reflected empirically in Fig. 5.3, where we consider the following experimental set-up in the classic FourRooms domain [24]. The agent is assumed to be given or have previously acquired four policies $\{\pi_0, \pi_1, \pi_2, \pi_3\}$ individually optimized to reach rewards located in each of the four rooms of the environment. There are three reward locations $\{g_0, g_1, g_2\}$ scattered across the rooms, each with its own initial reward and all with $\lambda = 0.5$. At the

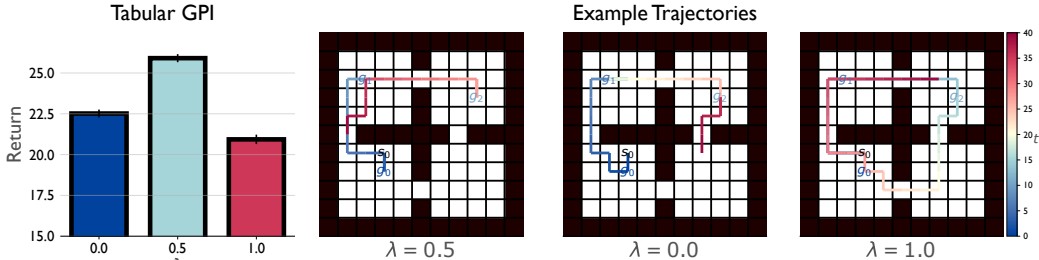

Figure 5.3: **Tabular GPI.** (Left) Average returns obtained by agents performing GPE+GPI using $\lambda$Rs with $\lambda \in \{0.0, 0.5, 1.0\}$ over 50 episodes. Error bars indicate standard error. (Right) Sample trajectories. Agents with $\lambda$ set too high overstay in rewarding states, and those with $\lambda$ too low leave too early.

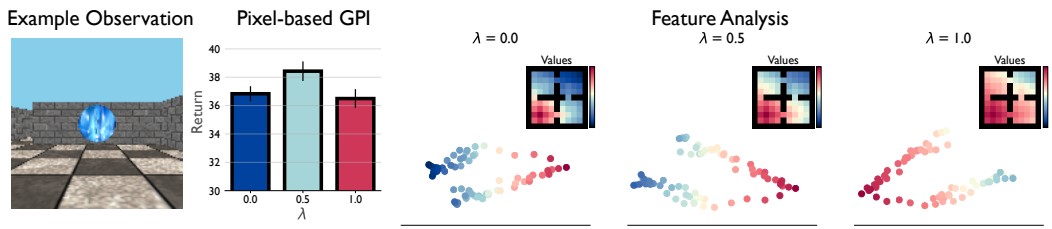

Figure 5.4: **Pixel-Based GPI.** Performance is strongest for agents using the correct $\lambda = 0.5$. PCA on the learned features in each underlying environment state shows that the $\lambda$Fs capture the value-conditioned structure of the environment.

beginning of each episode, an initial state $s_0$ is sampled uniformly from the set of available states. An episode terminates either when the maximum reward remaining in any of the goal states is less than $0.1$ or when the maximum number of steps $H = 40$ is reached (when $\lambda = 1$, the latter is the only applicable condition). For each of the four policies, we learn $\lambda$Rs with $\lambda \in \{0.0, 0.5, 1.0\}$ using dynamic programming and record the returns obtained while performing GPE+GPI with each of these representations over 50 episodes. Bellman error curves for the $\lambda$Rs are shown in Fig. F.3, and demonstrate that convergence is faster for lower $\lambda$. In the left panel of Fig. 5.3, we can indeed see that using the correct $\lambda$ (0.5) nets the highest returns. Example trajectories for each agent $\lambda$ are shown in the remaining panels.

**Pixel-Based Navigation**   We verified that the previous result is reflected in larger scales by repeating the experiment in a partially-observed version of FourRooms in which the agent receives $128 \times 128$ RGB egocentric observations of the environment (Fig. 5.4, left) with $H = 50$. In this case, the agent learns $\lambda$Fs for each policy for $\lambda \in \{0.0, 0.5, 1.0\}$, where each $\lambda$F is parameterized by a feedforward convolutional network with the last seven previous frames stacked to account for the partial observability. The base features were Laplacian eigenfunctions normalized to $[0, 1]$, which which were shown by Touati et al. [25] to perform the best of all base features for SFs across a range of environments including navigation.

## 6   Understanding Natural Behavior

Naturalistic environments often exhibit diminishing reward and give insight into animal behavior. The problem of foraging in an environment with multiple diminishing food patches (i.e., reward states) is of interest in behavioral science [26–28]. The cornerstone of foraging theory is the marginal value theorem [MVT; 28, 29], which states that the optimal time to leave a patch is when the patch's reward rate matches the average reward rate of the environment. However, the MVT does not describe which patch to move to once an agent leaves its current patch. We show that the $\lambda$O recovers MVT-like behavior in discrete environments and improves upon the MVT by not only predicting *when* agents should leave rewarding patches, but also *where* they should go. Moreover, we provide a scheme for *learning* $\lambda$ alongside the $\lambda$O using feedback from the environment.

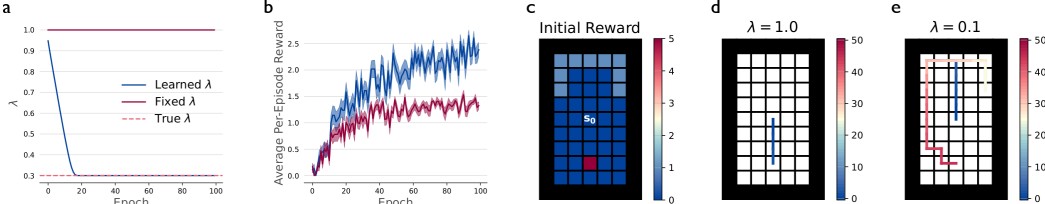

Figure 6.1: $\lambda$**O trained via FB. a)** $\lambda$ values of two agents in FourRooms, one which learns $\lambda$ and one which does not. **b)** Performance of the two agents from (a). Learning $\lambda$ improves performance. **c)** Reward structure and starting state of the asymmetric environment. **d)** Trajectory of an agent with $\lambda = 1$. The optimal strategy is to reach the high reward state and exploit it *ad infinitum.* **e)** Trajectory of an agent with $\lambda = 0.1$. The optimal strategy is to exploit each reward state for a few time steps before moving to the next reward state.

To learn the $\lambda$O, we take inspiration from the FBR [12] and use the following parametrization: $\Phi_\lambda^{\pi_z}(s, a, s') = F(s, a, z)^\top B(s')\mu(s')$ and $\pi_z(s) = \operatorname{argmax}_a F(s, a, z)^\top z$, where $\mu$ is a density with full support on $\mathcal{S}$, e.g., uniform. We then optimize using the the loss in Eq. (4.9) under this parameterization (details in Appendix E). Given a reward function $r : \mathcal{S} \to \mathbb{R}$ at evaluation time, the agent acts according to $\operatorname{argmax}_a F(s, a, z_R)^\top z_R$, where $z_R = \mathbb{E}_{s \sim \mu}[r(s)B(s)]$. Because the environment is non-stationary, $z_R$ has to be re-computed at every time step. To emulate a more realistic foraging task, the agent learns $\lambda$ by minimizing the loss in Eq. (H.1) in parallel with the loss

$$\mathcal{L}(\lambda) = \mathbb{E}_{s_t, s_{t+1} \sim \rho}\left[\mathbb{1}(s_t = s_{t+1})\left(r(s_{t+1}) - \lambda r(s_t)\right)^2\right],$$

which provably recovers the correct value of $\lambda$ provided that $\rho$ is sufficiently exploratory. In Appendix H.1 we provide experiments showing that using an incorrect value of $\lambda$ leads to poor performance on tabular tasks. In Fig. 6.1 we show that the agent learns the correct value of $\lambda$, increasing its performance. We illustrate the behavior of the $\lambda$O in an asymmetric environment that has one large reward state on one side and many small reward states (with higher total reward) on the other. Different values of $\lambda$ lead to very different optimal foraging strategies, which the $\lambda$O recovers and exhibits MVT-like behavior (see Appendix H.2 for a more detailed analysis). Our hope is that the $\lambda$R may provide a framework for new theoretical studies of foraging behavior and possibly mechanisms for posing new hypotheses. For example, an overly large $\lambda$ may lead to overstaying in depleted patches, a frequently observed phenomenon [30].

## 7 Conclusion

**Limitations** Despite its advantages, there are several drawbacks to the representation which are a direct result of the challenge of the DMU setting. First, the $\lambda$R is only useful for transfer across diminishing reward functions when the value of $\lambda$ at each state is consistent across tasks. In natural settings, this is fairly reasonable, as $\lambda$ can be thought of as encoding the type of the resource available at each state (i.e., each resource has its own associated decay rate). Second, as noted in Section 4.1, the $\lambda$R does not admit a measure-theoretic formulation, which makes it challenging to define a principled, occupancy-based version compatible with continuous state spaces. Third, the $\lambda$R is a prospective representation, and so while it is used to correctly evaluate a policy's future return under DMU, it is not inherently memory-based and so performs this evaluation as if the agent hasn't visited locations with diminishing reward before. Additional mechanisms (i.e., recurrence or frame-stacking) are necessary to account for previous visits. Finally, the $\lambda$R is dependent on an episodic task setting for rewards to reset, as otherwise the agent would eventually consume all reward in the environment. An even more natural reward structure would include a mechanism for reward *replenishment* in addition to depletion. We describe several such candidates in Appendix J, but leave a more thorough investigation of this issue to future work.

In this work, we aimed to lay the groundwork for understanding policy evaluation, learning, and composition under diminishing rewards. To solve such problems, we introduced—and showed the necessity of—the $\lambda$ *representation*, which generalizes the SR and FR. We demonstrated its usefulness for rapid policy evaluation and by extension, composition, as well as control. We believe the $\lambda$R represents a useful step in the development of state representations for naturalistic environments.

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

# Appendix: A State Representation for Diminishing Rewards

**GIFs of navigation agents can be found at** `lambdarepresentation.github.io` **and in the supplementary material.**

## A    Derivation of $\lambda$R Recursion

We provide a step-by-step derivation of the $\lambda$R recursion in Eq. (4.3):

$$
\begin{aligned}
\Phi_\lambda^\pi(s, s') &= \mathbb{E}\left[\sum_{k=0}^\infty \lambda^{n_t(s',k)}\gamma^k \mathbb{1}(s_{t+k} = s')\Big| s_t = s\right] \\
&= \mathbb{E}\left[\mathbb{1}(s_t = s') + \sum_{k=1}^\infty \lambda^{n_t(s',k)}\gamma^k \mathbb{1}(s_{t+k} = s')\Big| s_t = s\right] \\
&\overset{(i)}{=} \mathbb{E}\left[\mathbb{1}(s_t = s') + \lambda^{n_t(s',1)}\gamma \sum_{k=1}^\infty \lambda^{n_{t+1}(s',k)}\gamma^{k-1} \mathbb{1}(s_{t+k} = s')\Big| s_t = s\right] \\
&\overset{(ii)}{=} \mathbb{E}\Big[\mathbb{1}(s_t = s') + \mathbb{1}(s_t = s')\lambda\gamma \sum_{k=1}^\infty \lambda^{n_{t+1}(s',k)}\gamma^{k-1} \mathbb{1}(s_{t+k} = s') \\
&\qquad\qquad + \gamma(1 - \mathbb{1}(s_t = s'))\sum_{k=1}^\infty \lambda^{n_{t+1}(s',k)}\gamma^{k-1} \mathbb{1}(s_{t+k} = s')\Big| s_t = s\Big] \\
&= \mathbb{1}(s_t = s') + \mathbb{1}(s_t = s')\lambda\gamma \mathbb{E}_{s_{t+1}\sim p^\pi}\Phi_\lambda^\pi(s_{t+1}, s') \\
&\qquad\qquad + \gamma(1 - \mathbb{1}(s_t = s'))\mathbb{E}_{s_{t+1}\sim p^\pi}\Phi_\lambda^\pi(s_{t+1}, s') \\
&= \mathbb{1}(s_t = s')(1 + \gamma\lambda\mathbb{E}_{s_{t+1}\sim p^\pi}\Phi_\lambda^\pi(s_{t+1}, s')) \\
&\qquad\qquad + \gamma(1 - \mathbb{1}(s_t = s'))\mathbb{E}_{s_{t+1}\sim p^\pi}\Phi_\lambda^\pi(s_{t+1}, s'),
\end{aligned}
\tag{A.1}
$$

where $(i)$ is because $n_t(s', k) = n_t(s', 1) + n_{t+1}(s', k)$ and $(ii)$ is because

$$
\lambda^{n_t(s',1)} = \lambda^{\mathbb{1}(s_t=s')} = \mathbb{1}(s_t = s')\lambda + (1 - \mathbb{1}(s_t = s')).
$$

## B    Theoretical Analysis

Here, we provide proofs for the theoretical results in the main text.

**Lemma 3.1** (Impossibility; Informal)**.** *Given a reward function of the form Eq. (3.1), it is impossible to define a Bellman equation solely using the resulting value function and immediate reward.*

Before providing the formal statement and proof for Lemma 3.1, we introduce a definition for a Bellman operator.

**Definition B.1** (Bellman Operator)**.** *A Bellman operator is a contractive operator* $\mathbb{R}^{|\mathcal{S}|} \to \mathbb{R}^{|\mathcal{S}|}$ *that depends solely on* $\bar{\mathbf{r}}$*, one-step expectations under* $p^\pi$*, and learning hyperparameters (in our case* $\gamma$ *and* $\lambda$*).*

We can now give a formal statement of Lemma 3.1:

**Lemma B.1** (Impossibility; Formal). *Assume that $|\mathcal{S}| > 1$. Then, there does not exist a Bellman operator $T$ with fixed point $V^\pi$.*

*Proof.* Assume for a contradiction that $T$ is a Bellman operator. By the Banach fixed-point theorem, $V^\pi$ must be the unique fixed point of $T$. Hence, $TV^\pi$ must take on the following form (see the proof of Lemma 3.1 in Appendix B): for $s \in \mathcal{S}$,

$$(TV^\pi)(s) = \bar{r}(s) + \gamma\mathbb{E}_{s'\sim p^\pi(\cdot|s)}V^\pi(s') + \gamma(\lambda - 1)\bar{r}(s)\mathbb{E}_{s'\sim p^\pi(\cdot|s)}\Phi^\pi_\lambda(s', s).$$

For the assumption to hold, there must exist a function $f$ such that, for any $s \in \mathcal{S}$,

$$\mathbb{E}_{s'\sim p^\pi(\cdot|s)}\Phi^\pi_\lambda(s', s) = \mathbb{E}_{p^\pi(\cdot|s)}f(\bar{r}, V^\pi, \gamma, \lambda, s).$$

Now, by definition,

$$V^\pi(s) = \sum_{s'\in\mathcal{S}} \Phi^\pi_\lambda(s, s')\bar{r}(s').$$

$\bar{r}$ is a vector in $\mathbb{R}^{|\mathcal{S}|}$, so as long as $\mathcal{S} > 1$, $\bar{r}^\perp$ is non-trivial. Fix any $\bar{r}, V^\pi, s$. Pick a vector $\mathbf{w} \in \bar{r}^\perp \setminus \{\mathbf{0}\}$ and define

$$\tilde{\Phi}^\pi_\lambda(s, s') := \Phi^\pi_\lambda(s, s') + w(s')$$

for any $s, s' \in \mathcal{S}$. Note that

$$\sum_{s'\in\mathcal{S}} \tilde{\Phi}^\pi_\lambda(s, s')\bar{r}(s') = \sum_{s'\in\mathcal{S}} \Phi^\pi_\lambda(s, s')\bar{r}(s') + \sum_{s'\in\mathcal{S}} w(s')\bar{r}(s') = V^\pi(s),$$

as $\mathbf{w} \perp \bar{r}$. However,

$$\mathbb{E}_{s'\sim p^\pi(\cdot|s)}\tilde{\Phi}^\pi_\lambda(s', s) = \sum_{s'\in\mathcal{S}} p^\pi(s'|s)\Phi^\pi_\lambda(s', s) + \sum_{s'\in\mathcal{S}} p^\pi(s'|s)w(s).$$

The final term evaluates to $w(s)$. Because $\mathbf{w} \neq \mathbf{0}$, there must exist some $s$ such that $w(s) \neq 0$. For this $s$, we have a single input $(\bar{r}, V^\pi, \gamma, \lambda, s)$ to $f$ that corresponds to two distinct outputs: $\mathbb{E}_{s'\sim p^\pi(\cdot|s)}\Phi^\pi_\lambda(s', s)$ and $\mathbb{E}_{s'\sim p^\pi(\cdot|s)}\tilde{\Phi}^\pi_\lambda(s', s)$.

Hence, $f$ is a one-to-many mapping: for fixed input, there is more than one output. Therefore, $f$ is not a function, yielding a contradiction.

$\square$

The following establishes $\mathcal{G}^\pi_\lambda$ as a contraction.

**Lemma B.2** (Contraction). *Let $\mathcal{G}^\pi_\lambda$ be the operator as defined in Definition 4.2 for some stationary policy $\pi$. Then for any two matrices $\Phi, \Phi' \in \mathbb{R}^{|\mathcal{S}|\times|\mathcal{S}|}$,*

$$|\mathcal{G}^\pi_\lambda\Phi(s, s') - \mathcal{G}^\pi_\lambda\Phi'(s, s')| \leq \gamma|\Phi(s, s') - \Phi'(s, s')|.$$

*Proof.* We have

$$
\begin{aligned}
|(\mathcal{G}^\pi_\lambda\Phi - \mathcal{G}^\pi_\lambda\Phi')_{s,s'}| &= |(I \odot (\mathbf{1}\mathbf{1}^\mathsf{T} + \gamma\lambda P^\pi\Phi) + \gamma(\mathbf{1}\mathbf{1}^\mathsf{T} - I) \odot P^\pi\Phi \\
&\qquad - I \odot (\mathbf{1}\mathbf{1}^\mathsf{T} + \gamma\lambda P^\pi\Phi') - \gamma(\mathbf{1}\mathbf{1}^\mathsf{T} - I) \odot P^\pi\Phi')_{s,s'}| \\
&= |(I \odot \gamma\lambda P^\pi(\Phi - \Phi') + \gamma(\mathbf{1}\mathbf{1}^\mathsf{T} - I) \odot P^\pi(\Phi - \Phi'))_{s,s'}| \\
&= |((I \odot \lambda\mathbf{1}\mathbf{1}^\mathsf{T} + \mathbf{1}\mathbf{1}^\mathsf{T} - I) \odot \gamma P^\pi(\Phi - \Phi'))_{s,s'}| \\
&\overset{(i)}{\leq} |(\gamma P^\pi(\Phi - \Phi'))_{s,s'}| \\
&= \gamma|(P^\pi(\Phi - \Phi'))_{s,s'}| \\
&\leq \gamma|(\Phi - \Phi')_{s,s'}|,
\end{aligned}
$$

where $(i)$ comes from using $\lambda \leq 1$ and simplifying. $\qquad\square$

Note that we can actually get a tighter contraction factor of $\lambda\gamma$ for $s = s'$. Given this contractive property, we can prove its convergence with the use of the following lemma.

**Lemma B.3** (Max $\lambda$R). *The maximum possible value of $\Phi_\lambda^\pi(s, s')$ is*

$$\frac{\mathbb{1}(s = s') + (1 - \mathbb{1}(s = s'))\gamma}{1 - \lambda\gamma}.$$

*Proof.* For $s = s'$,

$$\Phi_\lambda^\pi(s, s) = 1 + \lambda\gamma\mathbb{E}_{s_{t+1}\sim p^\pi(\cdot|s_t)}\Phi_\lambda^\pi(s_{t+1}, s).$$

This is just the standard SR recursion with discount factor $\lambda\gamma$, so the maximum is

$$\sum_{k=0}^{\infty}(\lambda\gamma)^k = \frac{1}{1 - \lambda\gamma}. \tag{B.1}$$

For $s \neq s'$, $\mathbb{1}(s_t = s') = 0$, so

$$\Phi_\lambda^\pi(s, s') = \gamma\mathbb{E}_{s_{t+1}\sim p^\pi(\cdot|s_t)}\Phi_\lambda^\pi(s_{t+1}, s').$$

Observe that $\Phi_\lambda^\pi(s, s) \geq \Phi_\lambda^\pi(s, s')$ for $s' \neq s$, so the maximum is attained for $s_{t+1} = s'$. We can then use the result for $s = s'$ to get

$$\Phi_\lambda^\pi(s, s') = \gamma\left(\frac{1}{1 - \lambda\gamma}\right). \tag{B.2}$$

Combining Eq. (B.1) and Eq. (B.2) yields the desired result. $\qquad\square$

**Proposition 4.1** (Convergence). *Under the conditions assumed above, set $\Phi^{(0)} = (1 - \lambda)I$. For $k = 1, 2, \ldots$, suppose that $\Phi^{(k+1)} = \mathcal{G}_\lambda^\pi\Phi^{(k)}$. Then*

$$|(\Phi^{(k)} - \Phi_\lambda^\pi)_{s,s'}| \leq \frac{\gamma^{k+1}}{1 - \lambda\gamma}.$$

*Proof.* Using the notation $X_{s,s'} = X(s, s')$ for a matrix $X$:

$$\begin{aligned}
|(\Phi^{(k)} - \Phi_\lambda^\pi)_{s,s'}| &= |(\mathcal{G}_\lambda^k\Phi^{(0)} - \mathcal{G}_\lambda^k\Phi_\lambda^\pi)_{s,s'}| \\
&= |(\mathcal{G}_\lambda^k\Phi^{(0)} - \Phi_\lambda^\pi)_{s,s'}| \\
&\overset{(i)}{\leq} \gamma^k|(\Phi^{(0)} - \Phi_\lambda^\pi)_{s,s'}| \\
&\overset{(ii)}{=} \gamma^k\Phi_\lambda^\pi(s, s') \\
&\overset{(iii)}{\leq} \frac{\gamma^{k+1}}{1 - \lambda\gamma}
\end{aligned} \tag{B.3}$$

where $(i)$ is due to Lemma B.2, $(ii)$ is because $\Phi^{(0)}(s, s') = 0$ for $s \neq s'$, and $(iii)$ is due to Lemma B.3. $\qquad\square$

**Lemma B.4** (Subadditivity). *For any $s \in \mathcal{S}$, policy $\pi$, $\lambda \in [0, 1)$, and disjoint measurable sets $A, B \subseteq \mathcal{S}$,*

$$\Phi_\lambda^\pi(s, A \cup B) < \Phi_\lambda^\pi(s, A) + \Phi_\lambda^\pi(s, B).$$

*Proof.* Note that for disjoint sets $A, B$, we have $n_t(A \cup B, k) = n_t(A, k) + n_t(B, k)$. Hence, conditioned on some policy $\pi$ and $s_t = s$,

$$\lambda^{n_t(A \cup B, k)} \mathbb{P}(s_{t+k} \in A \cup B) = \lambda^{n_t(A,k)} \lambda^{n_t(B,k)} \mathbb{P}(s_{t+k} \in A) + \lambda^{n_t(A,k)} \lambda^{n_t(B,k)} \mathbb{P}(s_{t+k} \in B)$$
$$\leq \lambda^{n_t(A,k)} \mathbb{P}(s_{t+k} \in A) + \lambda^{n_t(B,k)} \mathbb{P}(s_{t+k} \in B),$$

where the first line follows from $\mathbb{P}(s_{t+k} \in A \cup B) = \mathbb{P}(s_{t+k} \in A) + \mathbb{P}(s_{t+k} \in B)$. Equality holds over all $A, B, t, k$ if and only if $\lambda = 1$. Summing over $k$ yields the result. $\qquad\square$

### B.1   Proof of Theorem 5.1

We first prove two results, which rely throughout on the fact that $\Phi_\lambda(s, a, s') \leq \frac{1}{1 - \lambda\gamma}$ for all $s, a, s'$, which follows from Lemma B.3. For simplicity, we also assume throughout that all rewards are non-negative, but this assumption can easily be dropped by taking absolute values of rewards. The proofs presented here borrow ideas from those of [16].

**Lemma B.5.** *Let $\{M_j\}_{j=1}^n \subseteq \mathcal{M}$ and $M \in \mathcal{M}$ be a set of tasks in an environment $\mathcal{M}$ with diminishing rate $\lambda$ and let $Q^{\pi_j^*}$ denote the action-value function of an optimal policy of $M_j$ when executed in $M$. Given estimates $\tilde{Q}^{\pi_j}$ such that $\|Q^{\pi_j^*} - \tilde{Q}^{\pi_j}\|_\infty \leq \epsilon$ for all $j$, define*

$$\pi(s) \in \operatorname*{argmax}_a \max_j \tilde{Q}^{\pi_j}(s, a).$$

*Then,*

$$Q^\pi(s, a) \geq \max_j Q^{\pi_j^*}(s, a) - \frac{1}{1 - \gamma} \left( 2\epsilon + \frac{\gamma(1 - \lambda)r(s, a)}{1 - \lambda\gamma} \right),$$

*where $r$ denotes the reward function of $M$.*

*Proof.* Define $\tilde{Q}_{\max}(s, a) := \max_j \tilde{Q}^{\pi_j}(s, a)$ and $Q_{\max}(s, a) := \max_j Q^{\pi_j^*}(s, a)$. Let $T^\nu$ denote the Bellman operator of a policy $\nu$ in task $M$. For all $(s, a) \in \mathcal{S} \times \mathcal{A}$ and

all $j$,

$$T_i^\pi \tilde{Q}_{\max}(s,a) = r(s,a) + \gamma \sum_{s'} p(s'|s,a) \left( (\lambda - 1) r_i(s,a) \Phi^\pi(s', \pi(s'), s) + \tilde{Q}_{\max}(s', \pi(s')) \right)$$

$$= r(s,a) + \gamma \sum_{s'} p(s'|s,a) \left( (\lambda - 1) r_i(s,a) \Phi^\pi(s', \pi(s'), s) + \max_b \tilde{Q}_{\max}(s', b) \right)$$

$$\geq r(s,a) + \gamma \sum_{s'} p(s'|s,a) \left( (\lambda - 1) r_i(s,a) \Phi^\pi(s', \pi(s'), s) + \max_b Q_{\max}(s', b) \right) - \gamma\epsilon$$

$$\geq r(s,a) + \gamma \sum_{s'} p(s'|s,a) \left( (\lambda - 1) r_i(s,a) \Phi^\pi(s', \pi(s'), s) + Q_{\max}(s', \pi_j^*(s')) \right) - \gamma\epsilon$$

$$\geq r(s,a) + \gamma \sum_{s'} p(s'|s,a) \left( (\lambda - 1) r_i(s,a) \Phi^\pi(s', \pi(s'), s) + Q_i^{\pi_j^*}(s', \pi_j^*(s')) \right) - \gamma\epsilon$$

$$= r(s,a) + \gamma \sum_{s'} p(s'|s,a) \left( (\lambda - 1) r_i(s,a) \Phi^{\pi_j^*}(s', \pi_j^*(s'), s) + Q_i^{\pi_j^*}(s', \pi_j^*(s')) \right) - \gamma\epsilon$$

$$+ \gamma(\lambda - 1) r(s,a) \sum_{s'} p(s'|s,a) \left( \Phi^\pi(s', \pi(s'), s) - \Phi^{\pi_j^*}(s', \pi_j^*(s'), s) \right)$$

$$\geq T_i^{\pi_j^*} Q_i^{\pi_j^*}(s,a) - \gamma\epsilon - \frac{\gamma(1-\lambda)r(s,a)}{1 - \lambda\gamma}$$

$$= Q_i^{\pi_j^*}(s,a) - \gamma\epsilon - \frac{\gamma(1-\lambda)r(s,a)}{1 - \lambda\gamma}.$$

This holds for any $j$, so

$$T^\pi \tilde{Q}_{\max}(s,a) \geq \max_j Q_i^{\pi_j^*}(s,a) - \gamma\epsilon - \frac{\gamma(1-\lambda)r(s,a)}{1 - \lambda\gamma}$$

$$= Q_{\max}(s,a) - \gamma\epsilon - \frac{\gamma(1-\lambda)r(s,a)}{1 - \lambda\gamma}$$

$$\geq \tilde{Q}_{\max}(s,a) - \epsilon - \gamma\epsilon - \frac{\gamma(1-\lambda)r(s,a)}{1 - \lambda\gamma}.$$

Next, note that for any $c \in \mathbb{R}$,

$$T^\pi (\tilde{Q}_{\max}(s,a) + c) = T^\pi \tilde{Q}_{\max}(s,a) + \gamma \sum_{s'} p(s'|s,a)c$$

$$= T^\pi \tilde{Q}_{\max}(s,a) + \gamma c.$$

Putting everything together, and using the fact that $T^\nu$ is monotonic and contractive,

$$Q_i^\pi(s, a) = \lim_{k \to \infty} (T^\pi)^k \tilde{Q}_{\max}(s, a)$$

$$\geq \lim_{k \to \infty} \left[ \tilde{Q}_{\max}(s, a) - \left( \epsilon(1 + \gamma) - \frac{\gamma(1 - \lambda)r(s, a)}{1 - \lambda\gamma} \right) \sum_{j=0}^{k} \gamma^j \right]$$

$$\geq \tilde{Q}_{\max}(s, a) - \frac{1}{1 - \gamma} \left( \epsilon(1 + \gamma) - \frac{\gamma(1 - \lambda)r(s, a)}{1 - \lambda\gamma} \right)$$

$$\geq Q_{\max}(s, a) - \epsilon - \frac{1}{1 - \gamma} \left( \epsilon(1 + \gamma) - \frac{\gamma(1 - \lambda)r(s, a)}{1 - \lambda\gamma} \right)$$

$$\geq Q^{\pi_j^*}(s, a) - \frac{1}{1 - \gamma} \left( 2\epsilon + \frac{\gamma(1 - \lambda)r(s, a)}{1 - \lambda\gamma} \right).$$

This holds for every $j$, hence the result. $\qquad\square$

**Lemma B.6.** *Let $\nu$ be any policy, $\lambda, \hat{\lambda} \in [0, 1]$, and $Q_\lambda$ denote a value function with respecting to diminishing rate $\lambda$. Then,*

$$\|Q_\lambda^\nu - Q_{\hat{\lambda}}^\nu\|_\infty \leq \frac{|\lambda - \hat{\lambda}|\|r\|_\infty}{1 - \gamma}.$$

*Proof.* The proof follows from the definition of $Q$: for every $(s, a) \in \mathcal{S} \times \mathcal{A}$,

$$|Q_\lambda^\nu(s, a) - Q_{\hat{\lambda}}^\nu(s, a)| = \left| \mathbb{E}_\pi \left[ \sum_{k=0}^{\infty} \gamma^k \left( \lambda^{n_t(s_{t+k}, k)} - \hat{\lambda}^{n_t(s_{t+k}, k)} \right) r(s_{t+k}) \Big| s_t = s, a_t = a \right] \right|$$

$$\leq \mathbb{E}_\pi \left[ \sum_{k=0}^{\infty} \gamma^k \left| \lambda^{n_t(s_{t+k}, k)} - \hat{\lambda}^{n_t(s_{t+k}, k)} \right| r(s_{t+k}) \Big| s_t = s, a_t = a \right]$$

$$= \mathbb{E}_\pi \left[ \sum_{k=0}^{\infty} \gamma^k r(s_{t+k}) \left| \lambda - \hat{\lambda} \right| \sum_{j=0}^{n_t(s_{t+k}, k)-1} \lambda^{n_t(s_{t+k}, k)-1-j} \hat{\lambda}^j \Big| s_t = s, a_t = a \right]$$

$$\leq |\lambda - \hat{\lambda}| \mathbb{E}_\pi \left[ \sum_{k=0}^{\infty} \gamma^k r(s_{t+k}) \Big| s_t = s, a_t = a \right]$$

$$\leq \frac{|\lambda - \hat{\lambda}|\|r\|_\infty}{1 - \gamma}.$$

$\qquad\square$

**Theorem 5.1** (GPI)**.** *Let $\{M_j\}_{j=1}^{n} \subseteq \mathcal{M}$ and $M \in \mathcal{M}$ be a set of tasks in an environment $\mathcal{M}$ and let $Q^{\pi_j^*}$ denote the action-value function of an optimal policy of $M_j$ when executed in $M$. Assume that the agent uses diminishing rate $\hat{\lambda}$ that may differ from the true environment diminishing rate $\lambda$. Given estimates $\tilde{Q}^{\pi_j}$ such that $\|Q^{\pi_j^*} - \tilde{Q}^{\pi_j}\|_\infty \leq \epsilon$ for all $j$, define*

$$\pi(s) \in \underset{a}{\arg\max} \max_{j} \tilde{Q}^{\pi_j}(s, a).$$

*Then,*

$$Q^\pi(s, a) \geq \max_{j} Q^{\pi_j^*}(s, a) - \frac{1}{1 - \gamma} \left( 2\epsilon + |\lambda - \hat{\lambda}|\|r\|_\infty + \frac{\gamma(1 - \lambda)r(s, a)}{1 - \lambda\gamma} \right).$$

*Proof.* Let $Q_\lambda$ denote a value function with respect to diminishing constant $\lambda$. We wish to bound

$$\max_j Q_{\hat{\lambda}}^{\pi_j^*}(s,a) - Q_\lambda^\pi(s,a),$$

i.e., the value of the GPI policy with respect to the true $\lambda$ compared to the maximum value of the constituent policies $\pi_j^*$ used for GPI, which were used assuming $\hat{\lambda}$. By the triangle inequality,

$$\max_j Q_{\hat{\lambda}}^{\pi_j^*}(s,a) - Q_\lambda^\pi(s,a) \leq \max_j Q_\lambda^{\pi_j^*}(s,a) - Q_\lambda^\pi(s,a) + |\max_j Q_\lambda^{\pi_j^*}(s,a) - \max_j Q_{\hat{\lambda}}^{\pi_j^*}(s,a)|$$

$$\leq \underbrace{\max_j Q_\lambda^{\pi_j^*}(s,a) - Q_\lambda^\pi(s,a)}_{(1)} + \max_j \underbrace{|Q_\lambda^{\pi_j^*}(s,a) - Q_{\hat{\lambda}}^{\pi_j^*}(s,a)|}_{(2)}.$$

We bound (1) by Lemma B.5 and (2) by Lemma B.6 to get the result. $\qquad\square$

### B.2 An Extension of Theorem 5.1

Inspired by [18], we prove an extension of Theorem 5.1:

**Theorem B.1.** *Let $M \in \mathcal{M}$ be a task in an environment $\mathcal{M}$ with true diminishing constant $\lambda$. Suppose we perform GPI assuming a diminishing constant $\hat{\lambda}$:*

> *Let $\{M_j\}_{j=1}^n$ and $M_i$ be tasks in $\mathcal{M}$ and let $Q_i^{\pi_j^*}$ denote the action-value function of an optimal policy of $M_j$ when executed in $M_i$. Given estimates $\tilde{Q}_i^{\pi_j}$ such that $\|Q_i^{\pi_j^*} - \tilde{Q}_i^{\pi_j}\|_\infty \leq \epsilon$ for all $j$, define $\pi(s) \in \operatorname{argmax}_a \max_j \tilde{Q}_i^{\pi_j}(s,a)$.*

*Let $Q_{\hat{\lambda}}^\pi$ and $Q_\lambda^{\pi^*}$ denote the action-value functions of $\pi$ and the $M$-optimal policy $\pi^*$ when executed in $M$, respectively. Then,*

$$\|Q_\lambda^{\pi^*} - Q_{\hat{\lambda}}^\pi\|_\infty \leq \frac{2}{1-\gamma}\left(\frac{1}{2}|\lambda - \hat{\lambda}|\|r\|_\infty + \epsilon + \|r - r_i\|_\infty + \min_j \|r_i - r_j\|_\infty\right) + \frac{1-\lambda}{1-\lambda\gamma}C,$$

*where $C$ is a positive constant not depending on $\lambda$:*

$$C = \gamma\frac{2\|r - r_i\|_\infty + 2\min_j \|r_i - r_j\|_\infty + \min\left(\|r\|_\infty, \|r_i\|_\infty\right) + \min\left(\|r_i\|_\infty, \|r_1\|_\infty, \ldots, \|r_n\|_\infty\right)}{1-\gamma}.$$

Note that when $\lambda = 1$, we recover Proposition 1 of [18] with an additional term quantifying error incurred by $\hat{\lambda} \neq \lambda$. The proof relies on two other technical lemmas, presented below.

**Lemma B.7.**

$$\|Q^{\pi^*} - Q_i^{\pi_i^*}\|_\infty \leq \frac{\|r - r_i\|_\infty}{1-\gamma} + \gamma(1-\lambda)\frac{\min\left(\|r\|_\infty, \|r_i\|_\infty\right) + \|r - r_i\|_\infty}{(1-\gamma)(1-\lambda\gamma)}.$$

*Proof.* Define $\Delta_i := \|Q^{\pi^*} - Q_i^{\pi_i^*}\|_\infty$. For any $(s, a) \in \mathcal{S} \times \mathcal{A}$,

$$
\begin{aligned}
|Q^{\pi^*}(s, a) - Q_i^{\pi_i^*}(s, a)| &= \Big| r(s, a) + \gamma \sum_{s'} p(s'|s, a) \left( (\lambda - 1) r(s, a) \Phi^{\pi^*}(s', \pi^*(s'), s) + Q^{\pi^*}(s', \pi^*(s')) \right) \\
&\quad - r_i(s, a) - \gamma \sum_{s'} p(s'|s, a) \left( (\lambda - 1) r_i(s, a) \Phi^{\pi_i^*}(s', \pi_i^*(s'), s) + Q_i^{\pi_i^*}(s', \pi_i^*(s')) \right) \Big| \\
&\leq |r(s, a) - r_i(s, a)| + \gamma \sum_{s'} p(s'|s, a) |Q^{\pi^*}(s, a) - Q_i^{\pi_i^*}(s, a)| \\
&\quad + \gamma(\lambda - 1) \sum_{s'} p(s'|s, a) \left| r(s, a) \Phi^{\pi^*}(s', \pi^*(s'), s) - r_i(s, a) \Phi^{\pi_i^*}(s', \pi_i^*(s'), s) \right| \\
&\leq \|r - r_i\|_\infty + \gamma \Delta_i + \gamma(1 - \lambda) \|r \Phi^{\pi^*} - r_i \Phi^{\pi_i^*}\|_\infty.
\end{aligned}
$$

The third term decomposes as

$$
\begin{aligned}
\|r \Phi^{\pi^*} - r_i \Phi^{\pi_i^*}\|_\infty &\leq \|r \Phi^{\pi^*} - r \Phi^{\pi_i^*}\|_\infty + \|r \Phi^{\pi_i^*} - r_i \Phi^{\pi_i^*}\|_\infty \\
&\leq \frac{\|r\|_\infty + \|r - r_i\|_\infty}{1 - \lambda\gamma}.
\end{aligned}
$$

We could equivalently use the following decomposition:

$$
\begin{aligned}
\|r \Phi^{\pi^*} - r_i \Phi^{\pi_i^*}\|_\infty &\leq \|r \Phi^{\pi^*} - r_i \Phi^{\pi^*}\|_\infty + \|r_i \Phi^{\pi^*} - r_i \Phi^{\pi_i^*}\|_\infty \\
&\leq \frac{\|r_i\|_\infty + \|r - r_i\|_\infty}{1 - \lambda\gamma},
\end{aligned}
$$

and so

$$
\|r \Phi^{\pi^*} - r_i \Phi^{\pi_i^*}\|_\infty \leq \frac{\min\left(\|r\|_\infty, \|r_i\|_\infty\right) + \|r - r_i\|_\infty}{1 - \lambda\gamma}.
$$

The inequalities above hold for all $s, a$ and so

$$
\begin{aligned}
\Delta_i &\leq \|r - r_i\|_\infty + \gamma \Delta_i + \gamma(1 - \lambda) \frac{\min\left(\|r\|_\infty, \|r_i\|_\infty\right) + \|r - r_i\|_\infty}{1 - \lambda\gamma} \\
\implies \Delta_i &\leq \frac{\|r - r_i\|_\infty}{1 - \gamma} + \gamma(1 - \lambda) \frac{\min\left(\|r\|_\infty, \|r_i\|_\infty\right) + \|r - r_i\|_\infty}{(1 - \gamma)(1 - \lambda\gamma)}.
\end{aligned}
$$

Hence the result. $\qquad\square$

**Lemma B.8.** *For any policy $\pi$,*

$$
\|Q_i^\pi - Q^\pi\|_\infty \leq \frac{\|r - r_i\|_\infty}{1 - \gamma} + \gamma(1 - \lambda) \frac{\|r - r_i\|_\infty}{(1 - \gamma)(1 - \lambda\gamma)}.
$$

*Proof.* Write $\Delta_i := \|Q_i^\pi - Q^\pi\|_\infty$. Proceeding as in the previous lemma, for all $(s,a) \in \mathcal{S} \times \mathcal{A}$, we have

$$|Q_i^\pi(s,a) - Q^\pi(s,a)| = \Big| r_i(s,a) + \gamma \sum_{s'} p(s'|s,a)\left((\lambda - 1)r_i(s,a)\Phi^\pi(s',\pi(s'),s) + Q_i^\pi(s',\pi(s'))\right)$$

$$- r(s,a) - \gamma \sum_{s'} p(s'|s,a)\left((\lambda - 1)r(s,a)\Phi^\pi(s',\pi(s'),s) + Q^\pi(s',\pi(s'))\right)$$

$$\leq |r(s,a) - r_i(s,a)| + \gamma \sum_{s'} p(s'|s,a)(1-\lambda)|r(s,a) - r_i(s,a)|\Phi^\pi(s',\pi(s'),s)$$

$$+ \gamma \sum_{s'} p(s'|s,a)|Q_i^\pi(s',\pi(s')) - Q^\pi(s',\pi(s'))|$$

$$\leq \|r - r_i\|_\infty + \gamma(1-\lambda)\|r - r_i\|_\infty \frac{1}{1 - \lambda\gamma} + \gamma\Delta_i'$$

$$\implies \Delta_i' \leq \|r - r_i\|_\infty + \frac{\gamma(1-\lambda)\|r - r_i\|_\infty}{1 - \lambda\gamma} + \gamma\Delta_i'$$

$$\implies \Delta_i' \leq \frac{\|r - r_i\|_\infty}{1 - \gamma} + \frac{\gamma(1-\lambda)\|r - r_i\|_\infty}{(1 - \gamma)(1 - \lambda\gamma)}.$$

$\square$

Finally, we prove Theorem B.1:

*Proof of Theorem B.1.* By the triangle inequality,

$$\|Q_\lambda^{\pi^*} - Q_{\hat{\lambda}}^\pi\|_\infty \leq \|Q_\lambda^{\pi^*} - Q_\lambda^\pi\|_\infty + \|Q_\lambda^\pi - Q_{\hat{\lambda}}^\pi\|_\infty.$$

By Lemma B.6, the second term is bounded above by

$$\frac{|\lambda - \hat{\lambda}|\|r\|_\infty}{1 - \gamma}.$$

The first term decomposes as follows (dropping the $\lambda$ subscript on all action-value functions for clarity):

$$\|Q^{\pi^*} - Q^\pi\|_\infty \leq \underbrace{\|Q^{\pi^*} - Q_i^{\pi_i^*}\|_\infty}_{(1)} + \underbrace{\|Q_i^{\pi_i^*} - Q_i^\pi\|_\infty}_{(2)} + \underbrace{\|Q_i^\pi - Q^\pi\|_\infty}_{(3)}.$$

Applying Lemma B.5 to (2) (but with respect to $M_i$ rather than $M$), we have that for any $j$,

$$Q_i^{\pi_i^*}(s,a) - Q_i^\pi(s,a) \leq Q_i^{\pi_i^*}(s,a) - Q_i^{\pi_j^*}(s,a) + \frac{1}{1 - \gamma}\left(2\epsilon + \frac{\gamma(1-\lambda)r_i(s,a)}{1 - \lambda\gamma}\right)$$

$$\implies \|Q_i^{\pi_i^*} - Q_i^\pi\|_\infty \leq \underbrace{\|Q_i^{\pi_i^*} - Q_j^{\pi_j^*}\|_\infty}_{(2.1)} + \underbrace{\|Q_j^{\pi_j^*} - Q_i^{\pi_j^*}\|_\infty}_{(2.2)} + \frac{1}{1 - \gamma}\left(2\epsilon + \frac{\gamma(1-\lambda)\|r_i\|_\infty}{1 - \lambda\gamma}\right).$$

We bound (2.1) using Lemma B.7 and (2.2) using Lemma B.8 (but with respect to $M_j$ rather than $M$):

$$\|Q_i^{\pi_i^*} - Q_j^{\pi_j^*}\|_\infty + \|Q_j^{\pi_j^*} - Q_i^{\pi_j^*}\|_\infty \leq \frac{2\|r_i - r_j\|_\infty}{1 - \gamma} + \gamma(1-\lambda)\frac{\min\left(\|r_i\|_\infty, \|r_j\|_\infty\right) + 2\|r_i - r_j\|_\infty}{(1 - \gamma)(1 - \lambda\gamma)}.$$

We then apply Lemma B.7 to (1) and Lemma B.8 to (3) to get the result.

$\square$

## C An $n$th Occupancy Representation

To generalize the first occupancy representation to account for reward functions of this type, it's natural to consider an $N$*th occupancy representation*—that is, one which accumulates value only for the first $N$ occupancies of one state $s'$ starting from another state $s$:

**Definition C.1** (NR). *For an MDP with finite $\mathcal{S}$, the $N$th-occupancy representation (NR) for a policy $\pi$ is given by $F^\pi \in [0, N]^{|\mathcal{S}| \times |\mathcal{S}|}$ such that*

$$\Phi^\pi_{(N)}(s, s') \triangleq \mathbb{E}_\pi \left[ \sum_{k=0}^\infty \gamma^{t+k} \mathbb{1}\left(s_{t+k} = s', \#\left(\{j \mid s_{t+j} = s', j \in [0, k-1]\}\right) < N\right) \Big| s_t \right]. \tag{C.1}$$

Intuitively, such a representation sums the first $N$ (discounted) occupancies of $s'$ from time $t$ to $t + k$ starting from $s_t = s$. We can also note that $\Phi^\pi_{(1)}$ is simply the FR and $\Phi_{(0)}(s, s') = 0 \; \forall s, s'$. As with the FR and the SR, we can derive a recursive relationship for the NR:

$$\Phi^\pi_{(N)}(s, s') = \mathbb{1}(s_t = s')(1 + \gamma \mathbb{E}\Phi^\pi_{(N-1)}(s_{t+1}, s')) + \gamma(1 - \mathbb{1}(s_t = s'))\mathbb{E}\Phi^\pi_{(N)}(s_{t+1}, s'), \tag{C.2}$$

where the expectation is wrt $p^\pi(s_{t+1}|s_t)$. Once again, we can confirm that this is consistent with the FR by noting that for $N = 1$, the NR recursion recovers the FR recursion. Crucially, we also recover the SR recursion in the limit as $N \to \infty$:

$$\lim_{N \mapsto \infty} \Phi^\pi_{(N)}(s, s') = \mathbb{1}(s_t = s')(1 + \gamma \mathbb{E}\Phi^\pi_{(\infty)}(s_{t+1}, s')) + \gamma(1 - \mathbb{1}(s_t = s'))\mathbb{E}\Phi^\pi_{(\infty)}(s_{t+1}, s')$$

$$= \mathbb{1}(s_t = s') + \gamma \mathbb{E}\Phi^\pi_{(\infty)}(s_{t+1}, s').$$

This is consistent with the intuition that the SR accumulates every (discounted) state occupancy in a potentially infinite time horizon of experience. While Definition C.1 admits a recursive form which is consistent with our intuition, Eq. (C.2) reveals an inconvenient intractability: the Bellman target for $\Phi^\pi_{(N)}$ requires the availability of $\Phi^\pi_{(N-1)}$. This is a challenge, because it means that if we'd like to learn any NR for finite $N > 1$, the agent also must learn and store $\Phi^\pi_{(1)}, \dots \Phi^\pi_{(N-1)}$. Given these challenges, the question of how to learn a tractable general occupancy representation remains. From a neuroscientific perspective, a fixed depletion amount is also inconsistent with both behavioral observations and neural imaging [3], which indicate instead that utility disappears at a fixed rate in proportion to the *current remaining utility*, rather than in proportion to the *original utility*. We address these theoretical and practical issues in the next section.

## D Additional Related Work

Another important and relevant sub-field of reinforcement learning is work which studies non-stationary rewards. Perhaps most relevant, the setting of DMU can be seen as a special case of submodular planning and reward structures [31, 32]. [31] focus specifically on planning and not the form of diminishment we study, while [32] is a concurrent work which focuses on the general class of submodular

reward problems and introduces a policy-based, REINFORCE-like method which is necessarily on-policy. In contrast, we focus on a particular sub-class of problems especially relevant to natural behavior and introduce a family of approaches which exploit this reward structure, are value-based (and which can be used to modify the critic in policy-based methods), and are compatible with off-policy learning. Other important areas include convex [33] and constrained [34, 35] MDPs. In these cases, non-stationarity is introduced by way of a primal-dual formulation of distinct problem classes into min-max games.

## E  Further Experimental Details

### E.1  Policy Evaluation

We perform policy evaluation for the policy shown in Fig. 4.1 on the $6 \times 6$ gridworld shown. The discount factor $\gamma$ was set to 0.9 for all experiments, which were run for $H = 10$ steps per episode. The error metric was the mean squared error:

$$Q_{error} \triangleq \frac{1}{|\mathcal{S}||\mathcal{A}|} \sum_{s,a} (Q^{\pi}(s, a) - \hat{Q}(s, a))^2, \tag{E.1}$$

where $Q^{\pi}$ is the ground truth $Q$-values and $\hat{Q}$ is the estimate. Transitions are deterministic. For the dynamic programming result, we learned the $\lambda \text{R}$ using Eq. (4.3) for $\lambda \in \{0.5, 1.0\}$ and then measured the resulting values by multiplying the resulting $\lambda \text{R}$ by the associated reward vector $\mathbf{r} \in \{-1, 0, 1\}^{36}$, which was $-1$ in all wall states and $+1$ at the reward state $g$. We compared the results to the ground truth values. Dynamic programming was run until the maximum Bellman error across state-action pairs reduced below 5e-2. For the tabular TD learning result, we ran the policy for three episodes starting from every available (non-wall) state in the environment, and learned the $\lambda \text{R}$ for $\lambda \in \{0.5, 1.0\}$ as above, but using the online TD update:

$\Phi_\lambda(s_t, a_t) \leftarrow \Phi_\lambda(s_t, a_t) + \alpha \delta_t,$

$\delta_t = \mathbf{1}(s_t) \odot (1 + \gamma \lambda \Phi_\lambda(s_{t+1}, a_{t+1})) + \gamma(1 - \mathbf{1}(s_t)) \odot \Phi_\lambda(s_{t+1}, a_{t+1}) - \Phi_\lambda(s_t, a_t),$

where $a_{t+1} \sim \pi(\cdot \mid s_{t+1})$. The learned $Q$-values were then computed in the same way as the dynamic programming case and compared to the ground truth. For the $\lambda \text{F}$ result, we first learned Laplacian eigenfunction base features as described in [25] from a uniform exploration policy and normalized them to the range $[0, 1]$. We parameterized the base feature network as a 2-layer MLP with ReLU activations and 16 units in the hidden layer. We then used the base features to learn the $\lambda \text{F}$s as in the tabular case, but with the $\lambda \text{F}$ network parameterized as a three-layer MLP with 16 units in each of the hidden layers and ReLU activations. All networks were optimized using Adam with a learning rate of 3e-4. The tabular and neural network experiments were repeated for three random seeds, the former was run for 1,500 episodes and the latter for 2,000.

### E.2  Policy Learning

We ran the experiments for Fig. 5.2 in a version of the TwoRooms environment from the NeuroNav benchmark [22] with reward modified to decay with a specified

---

Algorithm 1: Online Tabular $Q_\lambda$-Learning Update

---

1: **Require:** Current $\lambda$R-values $\Phi_\lambda^{(t)} \in \mathbb{R}^{|\mathcal{S}|\times|\mathcal{A}|\times|\mathcal{S}|}$, current reward vector $\mathbf{r}^{(t)}$, observed $(s_t, a_t, s_{t+1})$ tuple

2: Compute $Q_\lambda$-values: $Q_\lambda^{(t)} \leftarrow (\Phi_\lambda^{(t)})^\mathsf{T} \mathbf{r}^{(t)}$

3: Select greedy action: $a_{t+1} \leftarrow \mathrm{argmax}_{a\in\mathcal{A}} Q_\lambda^{(t)}(s_{t+1}, a)$

4: Update $\Phi_\lambda$:

$$\Phi_\lambda^{(t+1)}(s_t, a_t) \leftarrow \Phi_\lambda^{(t)}(s_t, a_t) + \alpha\delta^{(t)}, \quad \text{where}$$

$$\delta^{(t)} = \mathbf{1}(s_t) \odot (1 + \gamma\lambda\Phi_\lambda^{(t)}(s_{t+1}, a_{t+1})) + \gamma(1 - \mathbf{1}(s_t)) \odot \Phi_\lambda^{(t)}(s_{t+1}, a_{t+1}) - \Phi_\lambda^{(t)}(s_t, a_t).$$

5: **Return** updated $\Phi_\lambda^{(t+1)}$

---

$\lambda_{true} = 0.5$ and discount factor $\gamma = 0.95$. The initial rewards in the top right goal and the lower room goal locations were $5$ and the top left goal had initial reward $10$. The observations in the neural network experiment were one-hot state indicators. The tabular $Q_\lambda$ experiments run the algorithm in Algorithm 1 for 500 episodes for $\lambda \in \{0.0, 0.5, 1.0\}$, with $\lambda_{true}$ set to 0.5, repeated for three random seeds. Experiments used a constant step size $\alpha = 0.1$. There were five possible actions: up, right, down, left, and stay. The recurrent A2C agents were based on the implementation from the BSuite library [36] and were run for 7,500 episodes of maximum length $H = 100$ with $\gamma = 0.99$ using the Adam optimizer with learning rate 3e-4. The experiment was repeated for three random seeds. The RNN was an LSTM with 128 hidden units and three output heads: one for the policy, one for the value function, and one for the $\lambda$F. The base features were one-hot representations of the current state, 121-dimensional in this case.

### E.3  Tabular GPI

The agent is assumed to be given or have previously acquired four policies $\{\pi_0, \pi_1, \pi_2, \pi_3\}$ individually optimized to reach rewards located in each of the four rooms of the environment. There are three reward locations $\{g_0, g_1, g_2\}$ scattered across the rooms, each with its own initial reward $\bar{r} = [5, 10, 5]$ and all with $\lambda = 0.5$. At the beginning of each episode, an initial state $s_0$ is sampled uniformly from the set of available states. An episode terminates either when the maximum reward remaining in any of the goal states is less than $0.1$ or when the maximum number of steps $H = 40$ is reached. Empty states carry a reward of $0$, encountering a wall gives a reward of $-1$, and the discount factor is set to $\gamma = 0.97$.

For each of the four policies, we learn $\lambda$Rs with $\lambda$ equal to 0, 0.5, and 1.0 using standard dynamic programming (Bellman error curves plotted in Fig. F.3), and record the returns obtained while performing GPE+GPI with each of these representations over the course of 50 episodes. Bellman error curves for the $\lambda$Rs are In the left panel of Fig. 5.3, we can indeed see that using the correct $\lambda$ (0.5) nets the highest returns. Example trajectories for each of $\lambda$R are shown in the remaining panels.

### E.4 Pixel-Based GPI

In this case, the base policies $\Pi$ were identical to those used in the tabular GPI experiments. First, we collected a dataset consisting of 340 observation trajectories $(o_0, o_1, \ldots, o_{H-1}) \in \mathcal{O}^H$ with $H = 19$ from each policy, totalling $6,460$ observations. Raw observations were $128 \times 128 \times 3$ and were converted to grayscale. The previous seven observations were stacked and used to train a Laplacian eigenfunction base feature network in the same way as [25]. For observations less than seven steps from the start of an episode, the remaining frames were filled in as all black observations (i.e., zeros). The network consisted of four convolutional layers with 32 $3 \times 3$ filters with strides $(2, 2, 2, 1)$, each followed by a ReLU nonlinearity. This was then flattened and passed through a Layer Norm layer [37] and a $\mathtt{tanh}$ nonlinearity before three fully fully connected layers, the first two with 64 units each and ReLU nonlinearities and the final, output layer with 50 units. The output was $L_2$-normalized as in [25]. This network $\phi : \mathcal{O}^7 \mapsto \mathbb{R}^D$ (with $D = 50$) was trained on the stacked observations for 10 epochs using the Adam optimizer and learning rate 1e-4 with batch size $B = 64$. To perform policy evaluation, the resulting features, evaluated on the dataset of stacked observations were collected into their own dataset of $(s_t, a_{t+1}, s_{t+1}, a_{t+1})$ tuples, where $s_t \triangleq o_{t-6:t}$. The "states" were normalized to be between 0 and 1, and a vector $\mathbf{w}$ was fit to the actual associated rewards via linear regression on the complete dataset. The $\lambda$F network was then trained using a form of neural fitted Q-iteration [FQI; 38] modified for policy evaluation with $\lambda$Fs (Algorithm 2). The architecture for the $\lambda$F network was identical to the base feature network, with the exception that the hidden size of the fully connected layers was 128 and the output dimension was $D|\mathcal{A}| = 250$. FQI was run for $K = 20$ outer loop iterations, with each inner loop supervised learning setting run for $L = 100$ epochs on the current dataset. Supervised learning was done using Adam with learning rate 3e-4 and batch size $B = 64$. Given the trained networks, GPI proceeded as in the tabular case, i.e.,

$$a_t = \operatorname*{argmax}_{a \in \mathcal{A}} \max_{\pi \in \Pi} \mathbf{w}^\mathsf{T} \varphi_\theta^\pi(s_t, a). \tag{E.2}$$

50 episodes were run from random starting locations for $H = 50$ steps and the returns measured. Learning curves for the base features and for $\lambda$F fitting are shown in Fig. E.1. The $\lambda$F curve measures the mean squared error as in Eq. (E.1).

The feature visualizations were created by performing PCA to reduce the average $\lambda$F representations for observations at each state in the environment to 2D. Each point in the scatter plot represents the reduced representation on the $xy$ plane, and is colored according to the $\lambda$-conditioned value of the underlying state.

### E.5 Continuous Control

$\lambda$-**SAC**    See Appendix I for details.

### E.6 Learning the $\lambda$O with FB

Training the $\lambda$O with the FB parameterization proceeds in much the same way as in [12], but adjusted for a different norm and non-Markovian environment. We

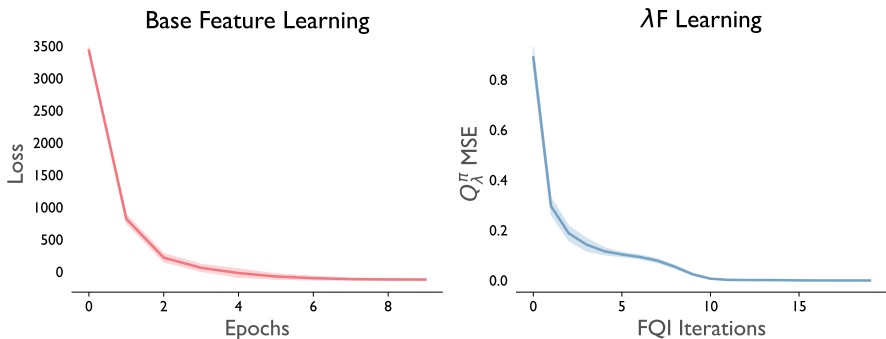

Figure E.1: **Learning curves for $\lambda$F policy evaluation.** Results are averaged over three runs, with shading indicating one unit of standard error.

---

Algorithm 2: Fitted $Q_\lambda$-Iteration

---

1: **Require**: Dataset of base features $\{\phi(s) \in \mathbb{R}^D\}_{s \in \mathcal{S}}$, decay rate $\lambda$, discount factor $\gamma$, reward feature vector $\mathbf{w} \in \mathbb{R}^D$, batch size $B$, learning rate $\alpha$
2: Initialize $\lambda$F $\varphi_\theta$ parameters $\theta^{(1)}$ (we drop the subscript $\lambda$ and superscript $\pi$ for concision)
3: **for** $k = 1 \ldots, K$ **do**
4:     // Stage 1: Construct dataset
5:     $\mathcal{D} \leftarrow \varnothing$
6:     **for** $(s, a) \in \mathcal{S} \times \mathcal{A}$ **do**
7:         **for** $(s', a') \in \mathcal{S} \times \mathcal{A}$ **do**
8: 
$$\mathcal{D} \leftarrow \mathcal{D} \cup \left\{ \left( (s, a), \underbrace{\mathbf{w}^\top \left[ \phi(s) \odot (1 + \lambda\gamma\bar{\varphi}_{\theta^{(k)}}(s', a')) + \gamma(1 - \phi(s)) \odot \bar{\varphi}_{\theta^{(k)}}(s', a') \right]}_{\triangleq y(s,a)} \right) \right\}$$
9:         **end for**
10:     **end for**
11:     // Stage 2: Supervised learning
12:     Randomly initialize $\theta_0$
13:     **for** $\ell = 1, \ldots, L$ **do**
14:         Randomly shuffle $\mathcal{D}$
15:         **for** $\{((s, a), y)\}_{b=1}^B \in \mathcal{D}$ **do**
16:             $\theta_\ell \leftarrow \theta_{\ell-1} - \alpha\nabla_\theta \frac{1}{2B} \sum_{b=1}^B \left( y_b - \mathbf{w}^\top\varphi_{\theta_{\ell-1}}(s_b, a_b) \right)^2$
17:         **end for**
18:     **end for**
19:     $\theta^{(k+1)} \leftarrow \theta_L$
20: **end for**

---

summarize the learning procedure in Algorithm 3. The loss function $\mathcal{L}$ is derived in Appendix H, with the addition of the following regularizer:

$$\left\| \mathbb{E}_{s \sim \rho} B_\omega(s) B_\omega(s)^\top - I \right\|^2.$$

This regularizer encourages $B$ to be approximately orthonormal, which promotes identifiability of $F_\theta$ and $B_\omega$ [12].

# F   Additional Results

See surrounding sections.

---

Algorithm 3: $\lambda$O FB Learning

---

1: **Require**: Probability distribution $\nu$ over $\mathbb{R}^d$, randomly initialized networks $F_\theta$, $B_\omega$, learning rate $\eta$, mini-batch size $B$, number of episodes $E$, number of epochs $M$, number of time steps per episode $T$, number of gradient steps $N$, regularization coefficient $\beta$, Polyak coefficient $\alpha$, initial diminishing constant $\lambda$, discount factor $\gamma$, exploratory policy greediness $\epsilon$, temperature $\tau$

2: // Stage 1: Unsupervised learning phase

3: $\mathcal{D} \leftarrow \varnothing$

4: **for** epoch $m = 1, \ldots, M$ **do**

5:    **for** episode $i = 1 \ldots, E$ **do**

6:       Sample $z \sim \nu$

7:       Observe initial state $s_1$

8:       **for** $t = 1, \ldots, T$ **do**

9:          Select $a_t$ $\epsilon-$greedy with respect to $F_\theta(s_t, a, z)^\top z$

10:         Observe reward $r_t(s_t)$ and next state $s_{t+1}$

11:         $\mathcal{D} \leftarrow \mathcal{D} \cup \{(s_t, a_t, r_t(s_t), s_{t+1})\}$

12:       **end for**

13:    **end for**

14:    **for** $n = 1, \ldots, N$ **do**

15:       Sample a minibatch $\{(s_j, a_j, r_j(s_j), s_{j+1})\}_{j \in J} \subset \mathcal{D}$ of size $|J| = B$

16:       Sample a minibatch $\{\tilde{s}_j\}_{j \in J} \subset \mathcal{D}$ of size $|\tilde{J}| = B$

17:       Sample a minibatch $\{s'_j\}_{j \in J} \overset{\text{iid}}{\sim} \mu$ of size $|J| = B$

18:       Sample a minibatch $\{z_j\}_{j \in J} \overset{\text{iid}}{\sim} \nu$ of size $|J| = B$

19:       For every $j \in J$, set $\pi_{z_j}(\cdot|s_{j+1}) = \texttt{softmax}\left(F_{\theta^-}(s_{j+1}, \cdot, z_j)^\top z_j / \tau\right)$

20:

$$
\begin{aligned}
\mathcal{L}(\theta, \omega) \leftarrow{} & \frac{1}{2B^2} \sum_{j,k \in J^2} \left( F_\theta(s_j, a_j, z_j)^\top B_\omega(s'_k) - \gamma \sum_{a \in \mathcal{A}} \pi_{z_j}(a|s_{j+1}) F_{\theta^-}(s_{j+1}, a, z_j)^\top B_{\omega^-}(s'_k) \right)^2 \\
& - \frac{1}{B} \sum_{j \in J} F_\theta(s_j, a_j, z_j)^\top B_\omega(s_j) \\
& + \frac{\gamma(1-\lambda)}{B} \sum_{j \in J} \mu(s_j) F_\theta(s_j, a_j, z_j)^\top B_\omega(s_j) \sum_{a \in \mathcal{A}} \pi_{z_j}(a|s_{j+1}) F_{\theta^-}(s_{j+1}, a, z_j)^\top B_{\omega^-}(s_j) \\
& + \beta \left( \frac{1}{B^2} \sum_{j,k \in J^2} B_\omega(s_j)^\top \bar{B}_\omega(\tilde{s}_k) \bar{B}_\omega(s_j)^\top \bar{B}_\omega(\tilde{s}_k) - \frac{1}{B} \sum_{j \in J} B_w(s_j)^\top \bar{B}_\omega(s_j) \right)
\end{aligned}
$$

21:       Update $\theta$ and $\omega$ via one step of $\texttt{Adam}$ on $\mathcal{L}$

22:       Sample a minibatch $\{(s_j, r_j(s_j), s_{j+1}, r_{j+1}(s_{j+1}))\}_{j \in J}$ of size $|J| = B$ from $\mathcal{D}$

23:       $\mathcal{L}_\lambda(\lambda) \leftarrow \frac{1}{2B} \sum_{j \in J} \mathbb{1}(s_{j+1} = s_j)\left(r_{j+1}(s_{j+1}) - \lambda r_j(s_j)\right)^2$

24:       Update $\lambda$ via one step of $\texttt{Adam}$ on $\mathcal{L}_\lambda$

25:    **end for**

26:    $\theta^- \leftarrow \alpha\theta^- + (1-\alpha)\theta$

27:    $\omega^- \leftarrow \alpha\omega^- + (1-\alpha)\omega$

28: **end for**

29: // Stage 2: Exploitation phase for a single episode with initial reward $r_0(s)$

30: $z_R \leftarrow \sum_{s \in \mathcal{S}} \mu(s) r_0(s) B_\omega(s)$

31: Observe initial state $s_1$

32: **for** $t = 1, \ldots, T$ **do**

33:    $a_t \leftarrow \text{argmax}_{a \in \mathcal{A}} F(s_t, a, z_R)^\top z_R$

34:    Observe reward $r_t(s)$ and next state $s_{t+1}$

35:    $z_R \leftarrow \sum_{s \in \mathcal{S}} \mu(s) r_t(s) B_\omega(s)$

36: **end for**

---

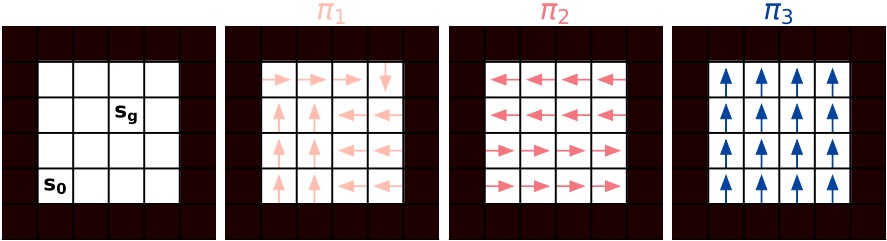

Figure F.1: **A simple grid and several policies.**

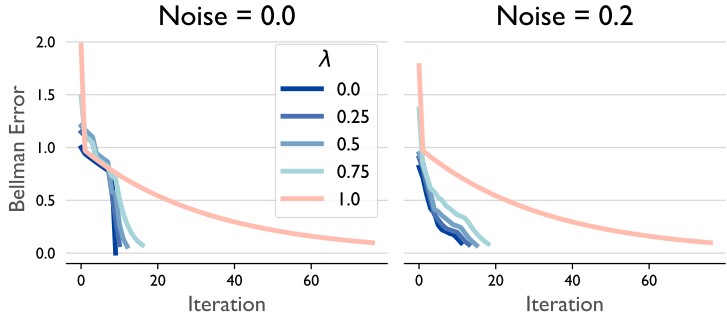

Figure F.3: **Convergence of dynamic programming on FourRooms with and without stochastic transitions.**

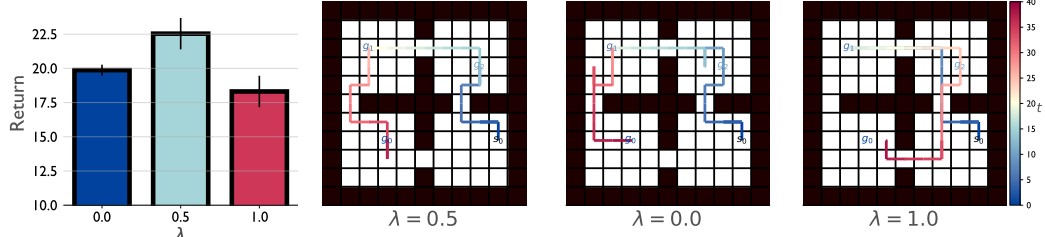

Figure F.4: **GPI with noisy transitions in FourRooms.** To verify that performance was maintained even with stochastic transitions, we added a 20% probability that a given action would result in a random transition to neighboring state. Results are consistent with Fig. 5.3, indicating the having the correct value of $\lambda$ produces better performance.

## G Advantage of the Correct $\lambda$

Importantly, for GPE using the $\lambda R$ to work in this setting, the agent must either learn or be provided with the updated reward vector $\mathbf{r}_\lambda$ after each step/encounter with a rewarded state. This is because the $\lambda R$ is forward-looking in that it measures the (diminished) expected occupancies of states in the future without an explicit mechanism for remembering previous visits. For simplicity in this case, we provide this vector to the agent at each step—though if we view such a multitask agent as simply as a module carrying out the directives of a higher-level module or policy within a hierarchical framework as in, e.g., Feudal RL [39], the explicit provision of reward information is not unrealistic. Regardless, a natural question in this case is whether there is actually any value in using the $\lambda R$ with the corret value of $\lambda$ in this

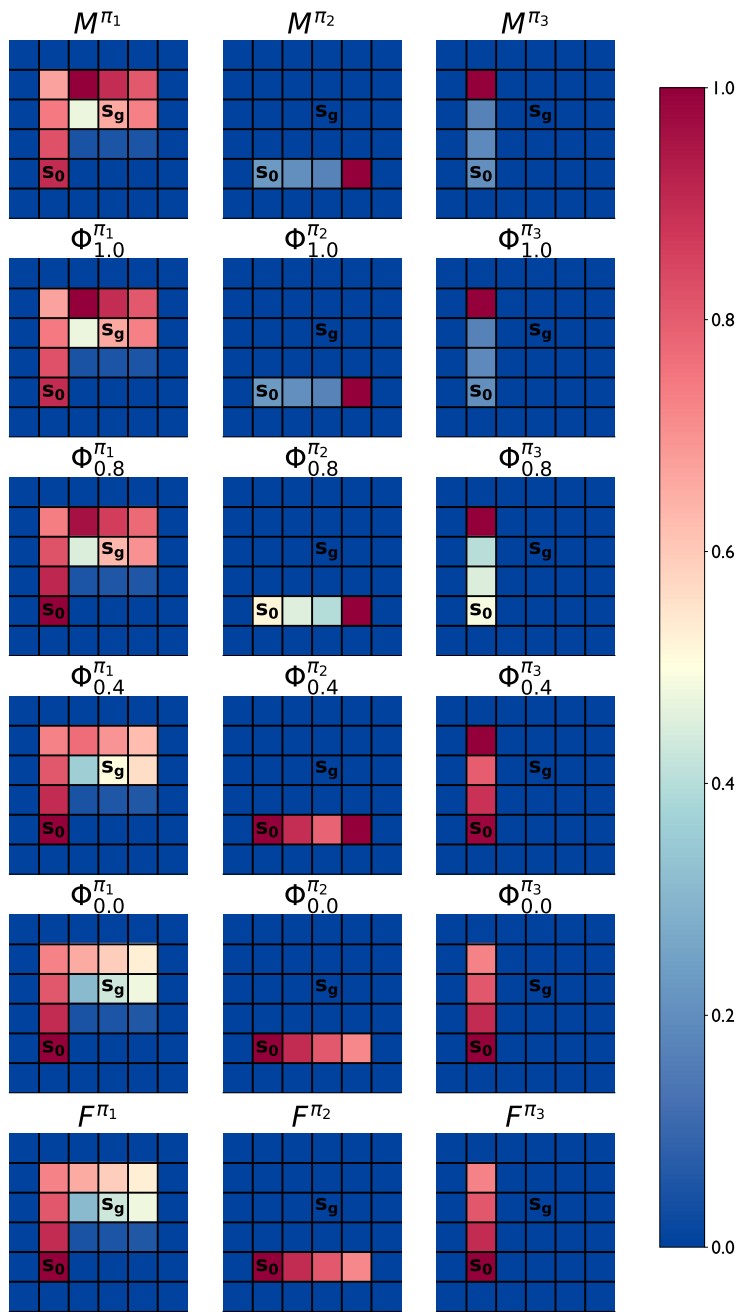

Figure F.2: **Visualizing the SR, the $\lambda$R and the FR**. We can see that the $\Phi_1^\pi$ is equivalent to the SR and $\Phi_0^\pi$ is equivalent to the FR, with intermediate values of $\lambda$ providing a smooth transition between the two.

setting: If the agent is provided with the correct reward vector, then wouldn't policy evaluation work with any $\lambda$R?

To see that this is not the case, consider the three-state toy MDP shown in Figure Fig. G.1, where $\bar{r}(s_1) = 10$, $\bar{r}(s_2) = 6$, $\bar{r}(s_0) = 0$, $\lambda(s_1) = 0$, $\lambda(s_2) = 1.0$, and $\gamma = 0.99$. At time $t = 0$, the agent starts in $s_0$. Performing policy evaluation with $\lambda(s_1) = \lambda(s_2) = 1$ (i.e., with the SR) would lead the agent to go left to $s_1$. However,

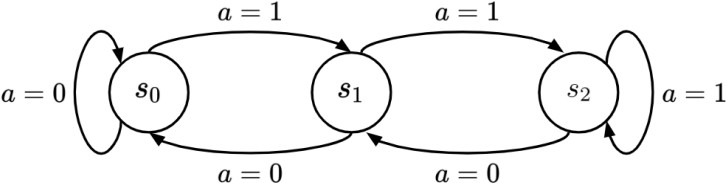

Figure G.1: **A 3-state toy environment**.

the reward would then disappear, and policy evaluation on the second step would lead it to then move right to $s_0$ and then $s_2$, where it would stay for the remainder of the episode. In contrast, performing PI with the correct values of $\lambda$ would lead the agent to go right to $s_2$ and stay there. In the first two timesteps, the first policy nets a total reward of $10 + 0 = 10$, while the second policy nets $6 + 5.94 = 11.94$. (The remaining decisions are identical between the two policies.) This is a clear example of the benefit of having the correct $\lambda$, as incorrect value estimation leads to suboptimal decisions even when the correct reward vector/function is provided at each step.

## H    The $\lambda$ Operator

To learn the $\lambda$O, we would like to define $\Phi_\lambda^\pi(s_t, ds') \triangleq \varphi_\lambda^\pi(s_t, s')\mu(ds')$ for some base policy $\mu$. However, this would lead to a contradiction:

$$\Phi_\lambda^\pi(s, A \cup B) = \int_A \varphi_\lambda^\pi(s, ds')\mu(ds') + \int_B \varphi_\lambda^\pi(s, ds')\mu(ds') = \Phi_\lambda^\pi(s, A) + \Phi_\lambda^\pi(s, B)$$

for all disjoint $A, B$, contradicting Lemma B.4.

For now, we describe how to learn the $\lambda$O for discrete $\mathcal{S}$, in which case we have $\Phi_\lambda^\pi(s, s') = \varphi_\lambda^\pi(s, s')\mu(s')$, i.e., by learning $\varphi$ we learn a weighted version of $\Phi$. We define the following norm, inspired by Touati et al. [25]:

$$\|\Phi_\lambda^\pi\|_\rho^2 \triangleq \mathbb{E}_{\substack{s \sim \rho \\ s' \sim \mu}}\left[\left(\frac{\Phi_\lambda^\pi(s, s')}{\mu(s')}\right)^2\right],$$

where $\mu$ is any density on $\mathcal{S}$. In the case of finite $\mathcal{S}$, we let $\mu$ be the uniform density. We then minimize the Bellman error for $\Phi_\lambda^\pi$ with respect to $\| \cdot \|_{\rho,\mu}^2$ (dropping the sub/superscripts on $\Phi$ and $\varphi$ for clarity):

$$\mathcal{L}(\Phi) = \|\varphi\mu - (I \odot (\mathbf{1}\mathbf{1}^{\mathsf{T}} + \lambda\gamma P^{\pi}\varphi\mu) + \gamma(\mathbf{1}\mathbf{1}^{\mathsf{T}} + I) \odot P^{\pi}\varphi\mu)\|_{\rho,\mu}^2$$

$$= \mathbb{E}_{s_t\sim\rho, s'\sim\mu}\Bigg[\bigg(\varphi(s_t, s') - \frac{\mathbb{1}(s_t = s')}{\mu(s')}$$
$$+ \gamma(1-\lambda)\frac{\mathbb{1}(s_t = s')}{\mu(s')}\mathbb{E}_{s_{t+1}\sim p^{\pi}(\cdot|s_t)}\bar{\Phi}(s_{t+1}, s') - \gamma\mathbb{E}_{s_{t+1}\sim p^{\pi}(\cdot|s_t)}\bar{\varphi}(s_{t+1}, s')\bigg)^2\Bigg]$$

$$\stackrel{+c}{=} \mathbb{E}_{s_t, s_{t+1}\sim\rho, s'\sim\mu}\Big[(\varphi(s_t, s') - \gamma\bar{\varphi}(s_{t+1}, s'))^2\Big]$$
$$- 2\mathbb{E}_{s_t, s_{t+1}\sim\rho}\Bigg[\sum_{s'}\mu(s')\varphi(s_t, s')\frac{\mathbb{1}(s_t = s')}{\mu(s')}\Bigg]$$
$$+ 2\gamma(1-\lambda)\mathbb{E}_{s_t, s_{t+1}\sim\rho}\Bigg[\sum_{s'}\mu(s')\varphi(s_t, s')\bar{\varphi}(s_{t+1}, s')\mu(s')\frac{\mathbb{1}(s_t = s')}{\mu(s')}\Bigg]$$

$$\stackrel{+c}{=} \mathbb{E}_{s_t, s_{t+1}\sim\rho, s'\sim\mu}\Big[(\varphi(s_t, s') - \gamma\bar{\varphi}(s_{t+1}, s'))^2\Big] - 2\mathbb{E}_{s_t\sim\rho}[\varphi(s_t, s_t)]$$
$$+ 2\gamma(1-\lambda)\mathbb{E}_{s_t, s_{t+1}\sim\rho}[\mu(s_t)\varphi(s_t, s_t)\bar{\varphi}(s_{t+1}, s_t)],$$

Note that we recover the SM loss when $\lambda = 1$. Also, an interesting interpretation is that when the agent can never return to its previous state (i.e., $\varphi(s_{t+1}, s_t) = 0$), then we also recover the SM loss, regardless of $\lambda$. In this way, the above loss appears to "correct" for repeated state visits so that the measure only reflects the first visit.

$$\mathcal{L}(\Phi) = \mathbb{E}_{s_t, a_t, s_{t+1}\sim\rho, s'\sim\mu}\Big[\big(F(s_t, a_t, z)^{\mathsf{T}}B(s') - \gamma\bar{F}(s_{t+1}, \pi_z(s_{t+1}), z)^{\mathsf{T}}\bar{B}(s')\big)^2\Big]$$
$$- 2\mathbb{E}_{s_t, a_t\sim\rho}\big[F(s_t, a_t, z)^{\mathsf{T}}B(s_t)\big]$$
$$+ 2\gamma(1-\lambda)\mathbb{E}_{s_t, a_t, s_{t+1}\sim\rho}\big[\mu(s_t)F(s_t, a_t, z)^{\mathsf{T}}B(s_t)\bar{F}(s_{t+1}, \pi_z(s_{t+1}), z)^{\mathsf{T}}\bar{B}(s_t)\big]$$
$$\tag{H.1}$$

Even though the $\lambda$O is not a measure, we can use the above loss to the continuous case, pretending as though we could take the Radon-Nikodym derivative $\frac{\Phi(s, ds')}{\mu(ds')}$.

## H.1 Experimental Results with the FB Parameterization

To show that knowing the correct value of $\lambda$ leads to improved performance, we trained $\lambda$O with the FB parameterization on the FourRooms task of Fig. 5.3, but with each episode initialized at a random start state and with two random goal states. Average per-epoch reward is shown in Fig. H.2. We tested performance with $\lambda_{\text{true}}, \lambda_{\text{agent}} \in \{0.5, 1.0\}$, where $\lambda_{\text{true}}$ denotes the true environment diminishing rate and $\lambda_{\text{agent}}$ denotes the diminishing rate that the agent uses. For the purpose of illustration, we do not allow the agent to learn $\lambda$. We see in Fig. H.2 that using the correct $\lambda$ leads to significantly increased performance. In particular, the left plot shows that assuming $\lambda = 1$, i.e., using the SR, in a diminishing environment can lead to highly suboptimal performance.

Hyperparameters used are given in Table 1 (notation as in Algorithm 3).

| Hyperparameter | Value |
|---|---|
| $M$ | 100 |
| $E$ | 100 |
| $N$ | 25 |
| $B$ | 128 |
| $T$ | 50 |
| $\gamma$ | 0.99 |
| $\alpha$ | 0.95 |
| $\eta$ | 0.001 |
| $\tau$ | 200 |
| $\epsilon$ | 1 |

Table 1: **λO-FB hyperparameters.**

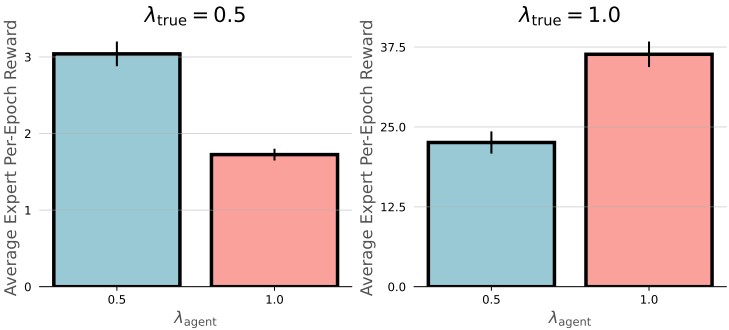

Figure H.1: **Performance of the λO-FB with two values of λ.** Results averaged over six seeds and 10 episodes per seed. Error bars indicate standard error.

## H.2    λO and the Marginal Value Theorem

To study whether the agent's behavior is similar to behavior predicted by the MVT, we use a very simple task with constant starting state and vary the distance between rewards (see Fig. H.1(a)). When an agent is in a reward state, we define an MVT-optimal leaving time as follows (similar to that of [8] but accounting for the non-stationarity of the reward).

Let $R$ denote the average per-episode reward received by a trained agent, $r(s_t)$ denote the reward received at time $t$ in a given episode, $R_t = \sum_{u=0}^{t} r(s_u)$ denote the total reward received until time $t$ in the episode, and let $T$ be episode length. Then, on average, the agent should leave its current reward state at time $t$ if the next reward that it would receive by staying in $s_t$, i.e., $\lambda r(s_t)$, is less than

$$\frac{R - R_t}{T}.$$

In other words, the agent should leave a reward state when its incoming reward falls below the diminished average per-step reward of the environment. We compute $R$ by averaging reward received by a trained agent over many episodes.

Previous studies have trained agents that assume stationary reward to perform foraging tasks, even though the reward in these tasks is non-stationary. These agents can still achieve good performance and MVT-like behavior [8]. However, because

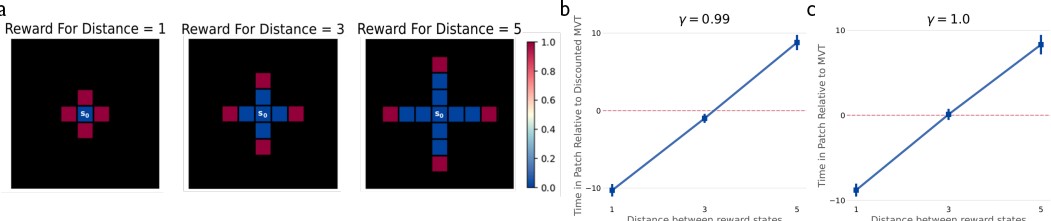

Figure H.2: **Analysis of MVT-like behavior of λO-FB. a)** Three environments with equal start state and structure but different distances between reward states. **b)** Difference between the agent's leave times and MVT-predicted leave times for $\gamma = 0.99$, with discounting taken into account. The agent on average behaves similar to the discounted MVT. **c)** Difference between the agent's leave times and MVT-predicted leave times for $\gamma = 1.0$, i.e., with no discounting taken into account. The agent on average behaves similar to the MVT. Results for (b) and (c) are averaged over three seeds. Error bars indicate standard error.

they target the standard RL objective

$$\mathbb{E}_\pi \left[ \sum_{k=0}^\infty \gamma^k r(s_{t+k}) \Big| s_t = s \right],$$

which requires $\gamma < 1$ for convergence, optimal behavior is recovered only with respect to the *discounted MVT*, in which $R$ (and in our case, $R_t$) weights rewards by powers of $\gamma$ [8].

In Fig. H.1(b-c) we perform a similar analysis to that of [8] and show that, on average over multiple distances between rewards, λO-FB performs similarly to the discounted MVT for $\gamma = 0.99$ and the standard MVT for $\gamma = 1.0$. An advantage of the λO is that it is finite for $\gamma = 1.0$ provided that $\lambda < 1$. Hence, as opposed to previous work, we can recover the standard MVT without the need to adjust for discounting.

Hyperparameters used are given in Table 1 (notation as in Algorithm 3).

## I SAC

**Mitigating Value Overestimation** One well-known challenge in deep RL is that the use of function approximation to compute values is prone to overestimation. Standard approaches to mitigate this issue typically do so by using *two* value functions and either taking the minimum $\min_{i \in \{1,2\}} Q_i^\pi(s, a)$ to form the Bellman target for a given $(s, a)$ pair [40] or combining them in other ways [41]. However, creating multiple networks is expensive in both computation and memory. Instead, we hypothesized that it might be possible to address this issue by using λ-based values. To test this idea, we modified the Soft Actor-Critic [SAC; 42] algorithm to compute λFs-based values by augmenting the soft value target $\mathcal{T}_{soft}Q = r_t + \gamma \mathbb{E}V_{soft}(s_{t+1})$, where $V_{soft}(s_{t+1})$ is given by the expression

$$\mathbb{E}_{a_{t+1} \sim \pi(\cdot|s_{t+1})} \Big[ \bar{Q}(s_{t+1}, a_{t+1}) + (\lambda - 1)\mathbf{w}^\mathsf{T}(\phi(s_t, a_t) \odot \varphi_\lambda(s_{t+1}, a_{t+1}))$$
$$- \alpha \log \pi(a_{t+1} \mid s_{t+1}) \Big]$$

A derivation as well as pseudocode for the modified loss is provided in Appendix E.5. Observe that for $\lambda = 1$, we recover the standard SAC value target, corresponding to an assumed stationary reward. We apply this modified SAC algorithm, which we term $\lambda$-SAC to feature-based Mujoco continuous control tasks within OpenAI Gym [43]. We found that concatenating the raw state and action observations $\tilde{\phi}_t = [s_t, a_t]$ and normalizing them to $[0, 1]$ make effective regressors to the reward. That is, we compute base features as

$$\phi_t^b = \frac{\tilde{\phi}_t^b - \min_b \tilde{\phi}_t^b}{\max_b \tilde{\phi}_t^b - \min_b \tilde{\phi}_t^b},$$

where $b$ indexes $(s_t, a_t)$ within a batch. Let $X \in [0, 1]^{B \times D}$ be the concatenated matrix of features for a batch, where $D = \dim(\mathcal{S}) + \dim(\mathcal{A})$. Then,

$$\mathbf{w}_t = \left(X^{\mathsf{T}} X\right)^{-1} X^{\mathsf{T}} \mathbf{r},$$

where here $\mathbf{r}$ denotes the vector of rewards from the batch. In addition to using a fixed $\lambda$ value, ideally we'd like an agent to adaptively update $\lambda$ to achieve the best balance of optimism and pessimism in its value estimates. Following [44], we frame this decision as a multi-armed bandit problem, discretizing $\lambda$ into three possible values $\{0, 0.5, 1.0\}$ representing the arms of the bandit. At the start of each episode, a random value of $\lambda$ is sampled from these arms and used in the value function update. The probability of each arm is updated using the Exponentially Weighted Average Forecasting algorithm [45], which modulates the probabilities in proportion to a feedback score. As in [44], we use the difference in cumulative (undiscounted) reward between the current episode $\ell$ and the previous one $\ell - 1$ as this feedback signal: $R_\ell - R_{\ell-1}$. That is, the probability of selecting a given value of $\lambda$ increases if performance is improving and decreases if it's decreasing. We use identical settings for the bandit algorithm as in [44]. We call this variant $\lambda$-SAC.

We plot the results for SAC with two critics (as is standard), SAC with one critic, SAC with a single critic trained with $\lambda$F-based values ("$x$-SAC" denotes SAC trained with a fixed $\lambda = x$), and $\lambda$-SAC trained on the HalfCheetah-v2 and Hopper-v2 Mujoco environments. All experiments were repeated over eight random seeds. HalfCheetah-v2 was found by [44] to support "optimistic" value estimates in that even without pessimism to reduce overestimation it was possible to perform well. Consistent with this, we found that single-critic SAC matched the performance of standard SAC, as did 1-SAC (which amounts to training a standard value function with the auxiliary task of SF prediction). Fixing lower values of $\lambda$ performed poorly, indicating that over-pessimism is harmful in this environment. However, $\lambda$-SAC eventually manages to learn to set $\lambda = 1$ and matches the final performance of the best fixed algorithms. Similarly, in [44] it was observed that strong performance in Hopper-v2 was associated with pessimistic value estimates. Consistent with this, $\lambda$-SAC learns to select lower values of $\lambda$, again matching the performance of SAC while only requiring one critic and significantly reducing the required FLOPS Fig. I.2. We consider these results to be very preliminary, and hope to perform more experiments on other environments. We also believe $\lambda$-SAC could be improved by using the difference between the current episode's total reward and the *average* of the total rewards from previous episodes $R_\ell - (\ell - 1)^{-1} \sum_{i=1}^{\ell-1} R_i$ as a more

stable feedback signal for the bandit. There is also non-stationarity in the base features due to the per-batch normalization, which could also likely be improved. Hyperparameters are described in Table 2.

| Hyperparameter | Value |
|---|---|
| Collection Steps | 1000 |
| Random Action Steps | 10000 |
| Network Hidden Layers | 256:256 |
| Learning Rate | $3 \times 10^{-4}$ |
| Optimizer | Adam |
| Replay Buffer Size | $1 \times 10^{6}$ |
| Action Limit | $[-1, 1]$ |
| Exponential Moving Avg. Parameters | $5 \times 10^{-3}$ |
| (Critic Update:Environment Step) Ratio | 1 |
| (Policy Update:Environment Step) Ratio | 1 |
| Has Target Policy? | No |
| Expected Entropy Target | $-\dim(\mathcal{A})$ |
| Policy Log-Variance Limits | $[-20, 2]$ |
| Bandit Learning Rate* | 0.1 |
| $\lambda$ Options* | $\{0, 0.5, 1.0\}$ |

Table 2: Hyperparameters for SAC Mujoco experiments. *Only applicable to $\lambda$-SAC

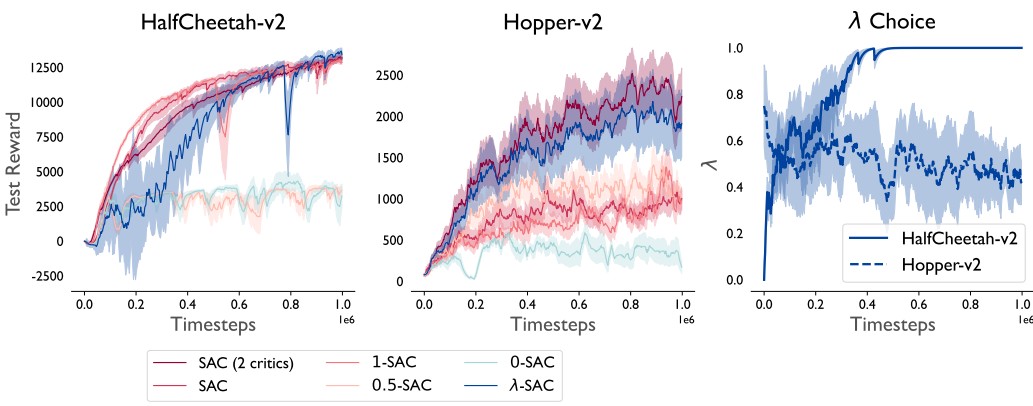

Figure I.1: $\lambda$-**SAC (1 critic) matches the performance of SAC (2 critics) by adapting $\lambda$ online.**

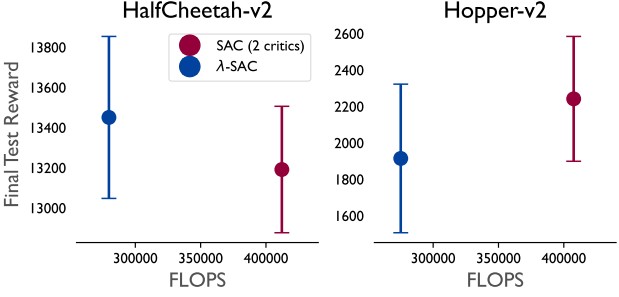

Figure I.2: $\lambda$-**SAC matches performance while saving in computational cost.**

## J  Replenishing Rewards

We list below a few candidate reward replenishment schemes, which are visualized in Fig. J.1.

**Time elapsed rewards**

$$r(s,t) = \lambda^{n(s,t)/m(s,t)}\bar{r}(s),$$

where $m(s,t)$ is the time elapsed since the last visit to state $s$:

$$m(s,t) \triangleq t - \max\{j|s_{t+j} = s, \, 0 \le j \le t-1\}.$$

Due to the $\max$ term in $m(s,t)$, the corresponding representation

$$\mathbb{E}_\pi\left[\sum_{k=0}^\infty \gamma^k \lambda^{n(s,t)/m(s,t)} \mathbb{1}(s_{t+k} = s')\Big| s_t = s\right]$$

does not admit a closed-form recursion. However, we empirically tested a version of this type of reward with $Q_\lambda$-learning in the TwoRooms environment, modified so that the exponent on $\lambda$ is $n(s,t)/(0.1m(s,t))$. This was done so that reward replenishes at a slow rate, reducing the deviation from the standard diminishing setting. Episode length was capped at $H = 100$. All other settings are identical to the $Q_\lambda$ experiment described in Appendix E. Results are presented in Fig. J.2 and a GIF is included in the supplementary material.

**Eligibility trace rewards**

$$r(s,t) = \left(1 - (1-\lambda_d)\sum_{j=0}^{t-1} \lambda_r^{t-j}\mathbb{1}(s_{t+j} = s)\right)\bar{r}(s),$$

where $\lambda_d, \lambda_r \in [0,1]$ are diminishment and replenishment constants, respectively. Denoting the corresponding representation by $\Omega^\pi$, i.e.,

$$\Omega^\pi(s,s') = \mathbb{E}\left[\sum_{k=0}^\infty \gamma^k\left(1 - (1-\lambda_d)\sum_{j=0}^k \lambda_r^{k-j}\mathbb{1}(s_{t+j} = s')\right)\mathbb{1}(s_{t+k} = s')\Big| s_t = s\right],$$

we obtain the following recursion:

$$\Omega^\pi(s,s') = \mathbb{1}(s = s')(\lambda_d - \gamma\lambda_r(1-\lambda_d)\mathbb{E}_{s_{t+1}\sim p^\pi(\cdot|s)}M_{\gamma\lambda_r}^\pi(s_{t+1}, s'))$$
$$+ \gamma\mathbb{E}_{s_{t+1}\sim p^\pi(\cdot|s)}\Omega^\pi(s_{t+1}, s'),$$

where $M_{\gamma\lambda_r}^\pi$ denotes the successor representation of $\pi$ with discount factor $\gamma\lambda_r$. This representation could be learned by alternating TD learning between $\Omega^\pi$ and $M_{\gamma\lambda_r}^\pi$. We leave this to future work.

**Total time rewards**

$$r(s,t) = \lambda_d^{n(s,t)}\lambda_r^{n(s,t)-t}\bar{r}(s),$$

where $\lambda_d, \lambda_r \in [0,1]$ are diminishment and replenishment constants, respectively. The corresponding representation is

$$P^\pi(s,s') = \mathbb{E}\left[\sum_{k=0}^\infty \gamma^k \lambda_d^{n_t(s',k)}\lambda_r^{k-n_t(s',k)}\mathbb{1}(s_{t+k} = s')\Big| s_t = s\right],$$

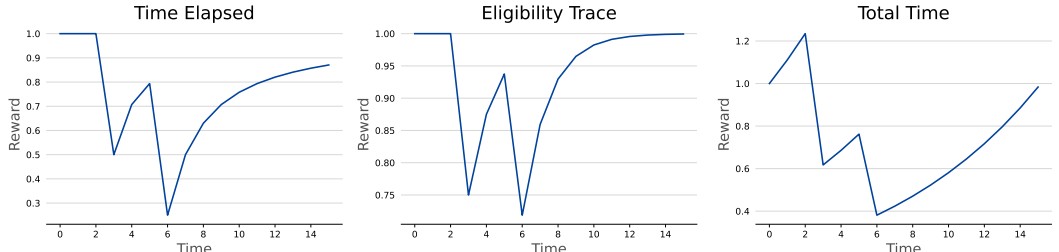

Figure J.1: **Visualizing three different replenishment schemes.** For all schemes, $\bar{r}(s) = 1$ and visits to $s$ are at $t = 2, 5$. (Left) The time elapsed reward with $\lambda = 0.5$; (Middle) The eligibility trace reward with $\lambda_r = \lambda_d = 0.5$; (Right) The total time reward with $\lambda_d = 0.5, \lambda_r = 0.9$.

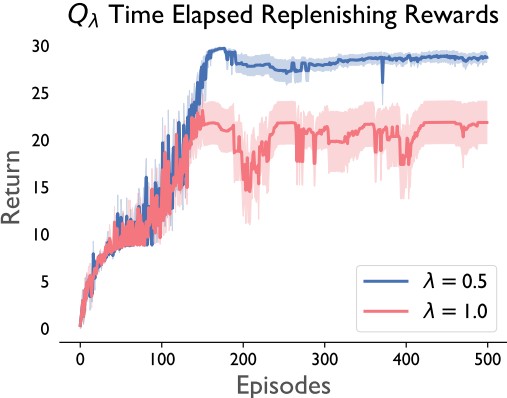

Figure J.2: **Performance on TwoRooms with replenishing rewards.** Return is averaged over five runs, with shading indicating one unit of standard error.

which satisfies the following recursion:

$$P^\pi(s, s') = \mathbb{1}(s = s') + \gamma(\lambda_d \mathbb{1}(s = s') + 1$$
$$- \mathbb{1}(s = s'))(\lambda_r(1 - \mathbb{1}(s = s')) + \mathbb{1}(s = s'))\mathbb{E}_{s_{t+1} \sim p^\pi(\cdot|s)} P^\pi(s_{t+1}, s').$$

While neither the reward nor the representation are guaranteed to be finite, $P^\pi$ could be learned via TD updates capped at a suitable upper bound.

## K  $\lambda$ vs. $\gamma$

We now briefly discuss the interaction between the temporal discount factor $\gamma$ commonly used in RL and the diminishing utility rate $\lambda$. The key distinction between the two is that all rewards decay in value every time step with respect to $\gamma$, regardless of whether a state is visited or not. With $\lambda$, however, decay is specific to each state (or $(s, a)$ pair) and only occurs when the agent visits that state. Thus, $\gamma$ decays reward in a global manner which is independent of the agent's behavior, and $\lambda$ decays reward in a local manner which dependent on the agent's behavior. In combination, they have the beneficial effect of accelerating convergence in dynamic programming (Fig. F.3). This indicates the potential for the use of higher discount factors in practice, as paired with a decay factor $\lambda$, similar (or faster) convergence rates could be observed even as agents are able to act with a longer effective temporal horizon.

## L   Compute Resources

The $\lambda$F-based experiments shown were run on a single NVIDIA GeForce GTX 1080 GPU. The recurrent A2C experiments took roughly 30 minutes, base feature learning for policy composition took approximately 45 minutes, $\lambda$F learning for policy composition took approximately 10 hours, and the SAC experiments took approximately 8 hours per run. The $\lambda$F training required roughly 30GB of memory due to the size of the dataset. All experiments in Section 6 and Appendix H were run on a single RTX5000 GPU and each training and evaluation run took about 30 minutes. All other experiments were run on a 2020 MacBook Air laptop 1.1 GHz Quad-Core Intel Core i5 CPU and took less than one hour to train.

## M   Broader Impact Statement

We consider this work to be primarily of a theoretical nature pertaining to sequential decision-making primarily in the context of natural intelligence. While it may have applications for more efficient training of artificial RL agents, it is hard to predict long-term societal impacts.

