# OpenReview forum: "A State Representation for Diminishing Rewards"
_NeurIPS.cc/2023/Conference — NeurIPS 2023 poster_

### Official Review · Reviewer_GnnZ · 2023-07-02

**Soundness:** 3 good
**Presentation:** 3 good
**Contribution:** 2 fair
**Rating:** 5
**Confidence:** 3

**Summary:**

The authors study a problem of DMU in RL, e.g. if visiting the same state may result in a smaller reward. They introduce a \lambda R representation that considers a particular form of diminishing rewards and provides convergence results. They extend it to continuous domains and perform preliminary experimental studies.

**Strengths:**

- Typically, such problems having non-markovian rewards are hard, but by having assumptions of \lamda rewards representation, the authors are able to recover some near-optimality convergence results.

- Extension to the continuous domain

**Weaknesses:**

I think the assumption of decay with \lambda is quite strong (very restrictive), and it's hard to imagine practical problems satisfying this. E.g. Authors have motivated the problem with examples of ice cream, finance and neuroscience, but it is hard(impossible) that a decay of visiting again is exactly a factor of 0<\lambda<1. Arguably a more interesting setting to consider would be:
- \lambda R for is a lower or upper bound on the true rewards that one may earn after visiting the states multiple times, i.e., the reward will decay at least by that amount or at max by a certain amount if visited again.
- I think another interesting setting will be if \lambda changes with state visitation frequency and thus needs to be estimated.

Both these settings will consider quite a large part of DMU-type problems.

Lemma 3.1 is perhaps not the right way to impose Impossibility/hardness results, and the proof needs to be rigorous. I think it can be interpreted in multiple ways, and also, the proof considers a specific format in which the reward is written and a specific definition of the value function. For the lemma statement to be true, the proof shall eradicate all the ways in which value function and rewards can be defined. For e.g. consider a reward function $r(s) = F(\tau \cup s) - F(\tau)$, where $F(\tau)$ is a general set function, $\tau$ is the agent's trajectory. $F(\tau)$ denotes the cumulative value earned for a trajectory $\tau$ as per the reward function (eqn 3.1). This can be written as Bellman recursion with $r(s_i) = F(s_{0:i}) - F(s_{0:i-1})$ (Imagine $\tau = \{s_0, s_1, s_2 ..... \}$). I would suggest not having it as a lemma but just in the text mentioning it is difficult to solve it with the Bellman equation directly.

Experiments: I think authors should consider more diverse reward functions, e.g., coverage functions, experiment design, and informative path planning objectives. For e.g., consider a camera-type measurement model, and if we observe the same region again, I will not get much information (there can be some noise model to establish \lamda type of decay). Similarly, in experiment design/ informative planning objectives: Information gained on the nearby states will decrease if I gather/collect measurements at my current location.

**Questions:**

Can you mention the assumptions behind Proposition 4.1? It is counterintuitive for me. $\lambda=0$ (FR) should be a much harder problem as compared to $\lamda=1$ (SR). However, error bound is better for $\lambda=0$?

I would suggest authors add an algorithm block for a high-level understanding of the algorithm.

I did not fully understand the experiments.
- Are your environments deterministic?
- In fig 5.2 b), why does \lambda =1 perform well? I would expect it to stay at one of the blue cells forever since it assumes that it gets the same reward always, and on the plot, it shall have an exponential decay.
- Is there an assumption of finite state-action spaces, or can you extend it to continuous dynamics as well?

Typos: line 268 the the, 156 when when,

**Limitations:**

no societal negative impact

---

> ### Author Rebuttal · Authors · 2023-08-09
>
> Thank you very much for your detailed review and helpful feedback! We are glad that you appreciated the convergence results and the extension of the $\lambda$R to continuous domains. We aim to address your concerns below:
>
> - Assumptions: This is a thought-provoking point! We completely agree that examining modifications of the specific DMU assumptions represent an interesting avenue for future work. We’d like to note that the neuroscience, foraging, and economics works studying DMU that we cite in the paper all use a fixed value for the diminishing rate (and in fact use the equivalent of a fixed value across states, an assumption which our framework can relax). Previous work [1,2] that applies standard RL methods to DMU problems also assumes a constant diminishment rate. In our work we show that it is possible to construct principled and tractable dynamic programming algorithms to solve DMU problems in RL and use these algorithms to achieve better performance than standard methods. Therefore, we believe that our work is an important first step towards RL problems with more general reward diminishment schemes, both theoretically and empirically. Additionally, we present the continuous control results in Attachment Figures L3 and L4 as an interesting example case where the “true” $\lambda$ is actually 1, but picking $\lambda < 1$ can actually result in a performance gain due to the tendency towards value overestimation in deep RL. In this case, $\lambda$-SAC estimates $\lambda$ online to maximize performance. More detail can be found in the general response above and in Appendix H.
>
>
> - Lemma 3.1: Thank you for alerting us to this! We agree that the lemma is unclear in its current form. Perhaps a more rigorous statement would be the following: “There does not exist a contractive operator $T^\pi$  that has fixed point $V^\pi$ and does not depend on $\Phi_\lambda^\pi$.” A sketch of the proof would be as follows: By the Banach fixed point theorem, $V^\pi$ is a unique fixed point, and so $T^\pi$ must be of the form given in the proof of Lemma 3.1. Hence, $T^\pi$ has a term containing $\Phi_\lambda^\pi$. For $T^\pi$ to not depend on $\Phi_\lambda^\pi$, we would need a function $f$ such that $\Phi_\lambda^\pi = f(\bar r, V^\pi)$. However, because $V^\pi = \bar r^\top \Phi_\lambda^\pi$, such a function cannot exist. In this sense the $\lambda$R is necessary for this problem setting, unlike the Markovian case in which using the SR is optional. We are happy to rewrite this as a remark rather than a lemma if you think it would be more appropriate. Regarding your concerns about our definition of reward and value being too narrow–as noted above, (1) our approach is consistent with studies of DMU in other fields, and (2) we believe theoretical results for specific settings are useful starting points for more general results in the future. Your suggestion to define reward as a difference of set-valued functions is interesting–thank you! However, can you please clarify your final point about writing a Bellman recursion for $r(s_i) = F(s_{0:i}) - F(s_{0:i-1})$?
>
> - Alternative Reward Functions: Thank you, this is an interesting point! In this paper, we’re primarily concerned with a specific setting as a starting point for studying DMU using RL. We believe a detailed examination of these alternative reward functions is beyond the scope of our work, but they represent an exciting opportunity for future study, and we will add a discussion to the paper.
>
> - Q1 (Proposition 4.1): Thank you for highlighting this! We agree that one of the most interesting, and perhaps broadly valuable, aspects of the $\lambda$R (and by extension, for $\lambda=0$, the FR) is that convergence is faster for lower values of $\lambda$. The assumptions are exactly those stated in the text, and the proofs can be found in Appendix B. We also show empirically in Figure E3/Attachment Figure L1 that dynamic programming indeed converges faster for lower $\lambda$. Intuitively, a lower $\lambda$ can be seen to speed convergence in a similar way to a lower discount factor in standard RL.
>
> - Q2 (Algorithm Block): Good point! Indeed, there are algorithm pseudocode blocks for $Q_\lambda$-Learning (Algorithm 1), Fitted $Q_\lambda$ Iteration (Algorithm 2), and $\lambda$O Forward-Backward Learning (Algorithm 3) in the Appendix, but can certainly add more details.
>
> - Q3 (Experiments): (i)  While the experiments presented in the initial submission all use deterministic environments, the theoretical analysis does not depend on this property, and we present new policy evaluation and composition results demonstrating that strong performance is maintained with stochastic dynamics in Attachment Figures L1 and L2. (ii) This is a subtle detail! It’s because, as we note in the conclusion, all of the representations we consider are prospective in nature and therefore don’t naturally account for previous visits. The $\lambda=1$ policy moves on because the amount of reward is decreasing–that is, $\vec r(s)$ is going down. The source of its suboptimality is that at every step using $\lambda = 1$ assumes that the current level of reward will persist, while using the correct $\lambda$ accounts for future decay. (iii) The $\lambda$F and $\lambda$O representations we introduce (Section 4.1, Appendix H, and Figures 5.4 and H.1) are designed to handle the continuous case.
>
> - Typos: Thank you for pointing these out! We’ll fix them.
>
> Thank you very much once again for your comments and questions! We will certainly integrate your suggestions into the paper. We hope our response has provided clarification and addressed your concerns. If that’s the case, we would greatly appreciate it if you would consider raising your score. If not, we would be more than happy to continue discussion!
>
> [1] https://openreview.net/forum?id=a0T3nOP9sB
>
> [2] https://proceedings.neurips.cc/paper/2020/hash/da97f65bd113e490a5fab20c4a69f586-Abstract.html

---

> > ### Comment · Reviewer_GnnZ · 2023-08-13
> > **Works that solve the same problem and a few follow up questions**
> >
> > Thank you for the response,
> >
> > I would like to bring the authors' attention towards non-markovian rewards in RL, which deals with a similar problem.
> > Submodularity is a known property of a set function and is an equivalent characterization of diminishing returns property. A good read for this topic could be  Krause A. and Golovin D., "Submodular Function Maximization". Chekuri and Pal, "A Recursive Greedy Algorithm for Walks in Directed Graphs" study planning on graphs providing both theoretical upper and lower bounds, and there are works which deal with planning under tabular MDP's under submodularity, Wang, et al. "Planning with Submodular Objective Functions" and, in fact also in the reinforcement learning setting with scalable policy gradient method, Prajapat, et al. Submodular reinforcement learning.
> >
> > There is also very related work which deals with convex RL where the objective is defined over state visitation distribution induced by the policy, Zahavy et al "Reward is enough for convex MDPs", general utilities RL, Kumar et al "Policy Gradient for Reinforcement Learning with General Utilities".
> >
> > A few follow-up questions:
> > - Can you please point me to a specific application and a section/line in the reference paper (or from neurology, economics) that deals with constant decay rate rewards?
> > - Regarding Proposition 4.1, Did I understand correctly that $\lambda=1$ corresponds to the usual RL setting where the rewards are fixed?  So $\lambda=0$ shall be a hard problem. In fact, there is quite some research demonstrating that hardness stems from a non-repeatable reward structure; please see Blum et al. Approximation Algorithms for Orienteering and Discounted-Reward TSP. I am not sure why you are getting a faster convergence for $\lambda=0$, while the TSP type of problems are known to be NP-hard. Do you also recover optimality guarantees for $\lambda=0$, or Maybe I missing something?
> > - Regarding Lemma 3.1, sorry, it is difficult to parse for me. Are you proving it by contradiction? If you are confident, you may have it as a proposition for being an independent, interesting result. (feel free to make multiple posts below to explain the proof below)
> >
> > Thanks,

---

> > > ### Author Response · Authors · 2023-08-14
> > > **Response Part 1/2**
> > >
> > > Regarding Non-Markovian Rewards: Thank you for the references. Indeed, there is a connection to submodular functions, which is partly what motivated our discussion of the $\lambda$ Operator (Section 4.1 and Appendix G). In particular, we note that submodularity is not as well-defined for continuous spaces, in that definitions which are equivalent in the finite case are no longer so in the infinite case. We appreciate the pointer to Wang, et al. "Planning with Submodular Objective Functions." It is certainly related, though differs in that it focuses solely on planning and not the specific form of diminishment we study. Similarly, we appreciate the pointer to Prajapat, et al. “Submodular Reinforcement Learning,” though we note that as it was only uploaded to arXiv a few weeks ago (after the submission date for this conference) we don’t think it affects the novelty of our work. However, we would be more than happy to include a discussion of this work as well. With respect to other examples of non-stationary RL, such as convex MDPs, we will absolutely include a discussion, as we have noted to Reviewer jk48. Something that we would like to emphasize is that, separate from DMU itself, we view the introduction of the $\lambda$R (and by extension, the $\lambda$F and $\lambda$O) as a useful contribution in that it unifies the SR, FR, and FB representations from the broader RL literature.
> > >
> > >
> > > Q1: Here is a list of several references with exponentially decaying utility (with the equivalent of constant $\lambda$), either in discrete or continuous time [1, 2, 3, 4, 5, 6]. One interesting note is that it is common in economics/decision theory papers to use one (constant) $\lambda$ for positive outcomes and a different $\lambda$ for negative outcomes. All of the rewards we study in our experiments are positive, so this specific situation does not directly apply in our settings, but extending our results to this setting would be an interesting direction for the future!
> > >
> > >
> > >
> > > Q2: Indeed, this is a subtle but important point. Proposition 4.1 is a convergence rate for policy evaluation under DMU (which includes the stationary case and the $\lambda=0$ case), not optimal control. Indeed, [7], which introduces the FR, describes the connection to TSP problems, and introduces a method “FR Planning” which finds provably shortest paths to a given goal given a fixed set of policies. In the case that there are multiple goals, this devolves to a TSP and is indeed NP-hard. However, computing the FR/$\lambda$R is more like computing the distance of a fixed route (policy) rather than finding the optimal policy. Thank you for bringing this up—we will emphasize it in the paper.
> > >
> > > [1] Pine, A., Seymour, B., Roiser, J. P., Bossaerts, P., Friston, K. J., Curran, H. V., & Dolan, R. J. (2009). Encoding of marginal utility across time in the human brain. Journal of Neuroscience, 29(30), 9575-9581. Figure 3b.
> > >
> > > [2] Wispinski, N., Butcher, A., Mathewson, K. W., Chapman, C. S., Botvinick, M., & Pilarski, P. M. (2022). Adaptive patch foraging in deep reinforcement learning agents. Transactions on Machine Learning Research. Figure 3c.
> > >
> > > [3] Chateauneuf, Alain, and Michèle Cohen. "Risk seeking with diminishing marginal utility in a non-expected utility model." Journal of Risk and Uncertainty 9 (1994): 77-91. Corollary 4.
> > >
> > > [4] Pratt, J. W. (1978). Risk aversion in the small and in the large. In Uncertainty in economics (pp. 59-79). Academic Press. Section 4.
> > >
> > > [5] Pine, A., Shiner, T., Seymour, B., & Dolan, R. J. (2010). Dopamine, time, and impulsivity in humans. Journal of Neuroscience, 30(26), 8888-8896. Eq. 2.
> > >
> > > [6] Rachlin, H. (1992). Diminishing marginal value as delay discounting. Journal of the Experimental Analysis of Behavior, 57(3), 407-415. Eq. 1.
> > >
> > > [7] Moskovitz, T., Wilson, S. R., & Sahani, M. (2021, October). A First-Occupancy Representation for Reinforcement Learning. In International Conference on Learning Representations.

---

> > > > ### Author Response · Authors · 2023-08-14
> > > > **Response Part 2/2**
> > > >
> > > > Q3:
> > > > **Definition:** A Bellman operator is a contractive operator $\\mathbb{R}^{|\\mathcal{S}|} \\rightarrow \\mathbb{R}^{|\\mathcal{S}|}$ that depends solely on $\\bar{r}$, one-step expectations under $p^\\pi$, and learning hyperparameters (in our case $\\gamma$ and $\\lambda$).
> > > >
> > > > For simplicity, below we assume the state-space $\\mathcal{S}$ is finite, and we let $\\mathbf{\\bar{r}}$ and $\\mathbf{V^\\pi}$ denote vectors of $\\bar r(s’)$ and $V^\\pi(s’)$, respectively, over $s’ \\in \\mathcal{S}$. Let $\\mathbf{v}^\\bot$ denote the orthogonal complement of a vector $\\mathbf{v}$.
> > > >
> > > > **Lemma 3.1 (revised):** Assume that $|\\mathcal{S}|>1$. Then, there does not exist a Bellman operator $T$ with fixed point $V^\\pi$.
> > > >
> > > > Proof: Assume for a contradiction that $T$ is a Bellman operator. By the Banach fixed-point theorem, $V^\\pi$ must be the unique fixed point of $T$. Hence, $TV^\\pi$ must take on the following form (see the proof of Lemma 3.1 in Appendix B): for $s \\in \\mathcal{S}$,
> > > >
> > > > $$(TV^\\pi)(s)= \\bar r(s) + \\gamma \\mathbb{E}\_{s’ \\sim p^\\pi(\\cdot|s)} V^\\pi(s’)  +\\gamma (\\lambda-1) \\bar r(s)  \\mathbb{E}\_{s’ \\sim p^\\pi(\\cdot|s)}\\Phi\_\\lambda^\\pi (s’,s).$$
> > > >
> > > > For the assumption to hold, there must exist a function $f$ (possibly depending on one-step expectations under $p^\\pi$) such that, for any $s \\in \\mathcal{S}$,
> > > >
> > > > $$\\mathbb{E}\_{s’ \\sim p^\\pi(\\cdot|s)} \\Phi\_\\lambda^\\pi(s’,s) =  f(\\mathbf{\\bar r}, \\mathbf{V^\\pi}, \\gamma, \\lambda, s).$$
> > > >
> > > > Now, by definition,
> > > >
> > > > $$
> > > > V^\\pi(s) = \\sum_{s’ \\in \\mathcal{S}} \\Phi_\\lambda^\\pi(s,s’) \\bar r(s’).
> > > > $$
> > > >
> > > > $\\mathbf{\\bar{r}}$ is a vector in $\\mathbb{R}^{|\\mathcal{S}|}$, so as long as $|\\mathcal{S}|>1$, $\\mathbf{\\bar{r}}^\\bot$ is non-trivial. Fix any $\\mathbf{\\bar{r}}, \\mathbf{V^\\pi}, s$. Pick a vector $\\mathbf{w} \\in \\mathbf{\\bar{r}}^\\bot \\setminus \\{\\mathbf{0}\\}$ and define
> > > >
> > > > $$
> > > > \\tilde{\\Phi}\_\\lambda^\\pi(s,s’) := \\Phi\_\\lambda^\\pi(s,s’) + w(s’)
> > > > $$
> > > >
> > > > for any $s,s’ \\in \\mathcal{S}$. Note that
> > > >
> > > > $$
> > > > \\sum\_{s’ \\in \\mathcal{S}} \\tilde{\\Phi}\_\\lambda^\\pi(s,s’) \\bar r(s’) = \\sum\_{s’ \\in \\mathcal{S}} \\Phi\_\\lambda^\\pi(s,s’) \\bar r(s’) +  \\sum\_{s’ \\in \\mathcal{S}} w(s’)\\bar r(s’) = V^\\pi(s),
> > > > $$
> > > >
> > > > as $\\mathbf{w} \\perp \\mathbf{\\bar{r}}$. However,
> > > >
> > > > $$
> > > > \\mathbb{E}\_{s’ \\sim p^\\pi(\\cdot|s)} \\tilde{\\Phi}\_\\lambda^\\pi(s’,s) = \\sum\_{s’ \\in \\mathcal{S}} p^\\pi(s’|s)  \\Phi\_\\lambda^\\pi(s’,s) + \\sum\_{s’ \\in \\mathcal{S}} p^\\pi(s’|s) w(s).
> > > > $$
> > > >
> > > > The final term evaluates to $w(s)$. Because $\\mathbf{w} \\neq \\mathbf{0}$, there must exist some $s$ such that $w(s) \\neq 0$. For this $s$, we have a single input $(\\mathbf{\\bar r}, \\mathbf{V^\\pi}, \\gamma, \\lambda, s)$ to $f$ that corresponds to two distinct outputs: $\\mathbb{E}\_{s’ \\sim p^\\pi(\\cdot|s)} \\Phi\_\\lambda^\\pi(s’,s)$ and $\\mathbb{E}\_{s’ \\sim p^\\pi(\\cdot|s)} \\tilde{\\Phi}\_\\lambda^\\pi(s’,s)$.
> > > >
> > > > Hence, $f$ is a one-to-many mapping: for fixed input, there is more than one output. Therefore, $f$ is not a function, yielding a contradiction.

---

> > > > > ### Author Response · Authors · 2023-08-16
> > > > > **Brief Addendum**
> > > > >
> > > > > Thank you once again for your feedback! We would just like to briefly add to our response regarding the concurrent work on submodular RL. While it is certainly true that diminishing rewards can be characterized as a type of submodular reward, we believe the contributions of these works are complementary.
> > > > >
> > > > > Prajapat, et al. focus on this general class of problems and introduce a policy-based, REINFORCE-like method which, while targeted at a general class of problems is on-policy and may suffer from high variance. In contrast, we focus on a particular sub-class of problems especially relevant to natural behavior and introduce a family of approaches which exploit this reward structure, are value-based (and which can be used to modify the critic in policy-based methods), and are compatible with off-policy learning. In short, while this concurrent work is focused on a more general setting, we consider a specific (but we believe especially interesting) instance of this problem class to derive a  solution that directly exploits this structure. In the process, we show that it's possible unify several state representations from the literature into the $\lambda$R. We think both perspectives have value.
> > > > >
> > > > > Thank you once again for your time and consideration.

---

> > > > > > ### Comment · Reviewer_GnnZ · 2023-08-18
> > > > > > **Response to Authors**
> > > > > >
> > > > > > Thank you for the proof and the clarification.
> > > > > >
> > > > > > I am still not able to convince myself of how do you get around the hardness of the problem. If I understood the reply, the authors are saying that they are not converging to the optimal policy but some state representation? Does it correspond to local convergence, but that defeats the Bellman type operator's purpose, doesn't it?
> > > > > >
> > > > > > But theorem 5.1 is, in fact, Guaranteeing global governance with some additive error. One possible explanation I can think of is if line 231 has $\lambda=0$, then the additive error for $\gamma~1$ is actually larger than the $Q$ function and thus, the bound is not meaningful anymore.
> > > > > >
> > > > > > A minor thing: eqn 2.4 shall have $s_t =s$

---

> > > > > > > ### Author Response · Authors · 2023-08-19
> > > > > > > **Response to Reviewer**
> > > > > > >
> > > > > > > Thank you for your response! Indeed, we are not attempting to claim a fast convergence rate to the optimal policy in a DMU setting.
> > > > > > >
> > > > > > > - Proposition 4.1 gives the convergence rate for learning the lambda representation associated with a *fixed* policy $\pi$.  This representation can then be used to evaluate $\pi$ (i.e. compute $V^\pi$) for any DMU-type task with non-Markov reward scheme defined in the same environment.
> > > > > > >
> > > > > > > - Proposition 5.1 is a generalization of the GPI result of Barreto et al. (2019) [1]. It is not about convergence, and shows improvement over a set of fixed policies rather than optimality.  GPI is a mechanism for composing *previously learned* policies, that are each near-optimal for a different task $M_j$.  The result then gives a performance bound for the composed policy on a new task $M$, as a function of how close to optimal the old policies are for their respective tasks, as well as how close the $\lambda$ value used to evaluate these policies is from the true decay rate.  None of these policies are necessarily optimal for the new task $M$.  For additional work on GPI, we refer to [2, 3, 4, 5, 6].
> > > > > > >
> > > > > > > We provide proofs for both results in Appendix B (as well as several additional results), and would be happy to discuss any details that are unclear.
> > > > > > >
> > > > > > > Re: Eq. 2.4 -- We actually explain this notation (using $\mathbb E[\cdot|s_t]$ as shorthand for $\mathbb E[\cdot|s_t=s]$) in Section 2, Lines 66-68, but we agree it could be confusing! We can adjust it.
> > > > > > >
> > > > > > > Thank you once again for the helpful discussion, and please let us know if anything remains unclear!
> > > > > > >
> > > > > > >
> > > > > > > [1] Barreto, A., Borsa, D., Quan, J., Schaul, T., Silver, D., Hessel, M., ... & Munos, R. (2018, July). Transfer in deep reinforcement learning using successor features and generalised policy improvement. In International Conference on Machine Learning (pp. 501-510). PMLR.
> > > > > > >
> > > > > > > [2] Barreto, A., Dabney, W., Munos, R., Hunt, J. J., Schaul, T., van Hasselt, H. P., & Silver, D. (2017). Successor features for transfer in reinforcement learning. Advances in neural information processing systems, 30.
> > > > > > >
> > > > > > > [3]  Thakoor, S., Rowland, M., Borsa, D., Dabney, W., Munos, R., & Barreto, A. (2022, June). Generalised policy improvement with geometric policy composition. In International Conference on Machine Learning (pp. 21272-21307). PMLR.
> > > > > > >
> > > > > > > [4] Barreto, A., Hou, S., Borsa, D., Silver, D., & Precup, D. (2020). Fast reinforcement learning with generalized policy updates. Proceedings of the National Academy of Sciences, 117(48), 30079-30087.
> > > > > > >
> > > > > > > [5] Borsa, D., Barreto, A., Quan, J., Mankowitz, D. J., van Hasselt, H., Munos, R., ... & Schaul, T. (2018, September). Universal Successor Features Approximators. In International Conference on Learning Representations.
> > > > > > >
> > > > > > > [6] Nemecek, M., & Parr, R. (2021, July). Policy caches with successor features. In International Conference on Machine Learning (pp. 8025-8033). PMLR.

---

> > > > > > > > ### Comment · Reviewer_GnnZ · 2023-08-20
> > > > > > > > **Response to Authors**
> > > > > > > >
> > > > > > > > Thanks for the reply. I have no more pending questions.

---

> > > > > > > > > ### Author Response · Authors · 2023-08-21
> > > > > > > > > **Thank you!**
> > > > > > > > >
> > > > > > > > > Thank you very much for your engagement and feedback--we will certainly integrate it into the paper!

---

### Official Review · Reviewer_2Ebj · 2023-07-05

**Soundness:** 3 good
**Presentation:** 3 good
**Contribution:** 2 fair
**Rating:** 6
**Confidence:** 3

**Summary:**

This paper studies the phenomenon of diminishing marginal utility in RL, where the a state-based reward r(s) decays as the agent visits it more often. To solve this problem setting, this work introduced a novel state representation, named the λ representation (λR). The author showed that we fail to define a Bellman equation in the λR setting and introduced a recursive relationship.

**Strengths:**

- The probelm setting of DMU in RL is interesting.
- The writing is clear and easy to follow.
- The paper provides theoretical analysis of the λ representations.

**Weaknesses:**

- Typos and grammar error: line 156 (when when ...), line 156 (Instead of keeping ...), line 177 (The mechanism)
- Missing baselines: the current experiments do not compare to other SOTA exploration baselines with intrinsic rewards. So it's hard to evaluate the effectiveness of the proposed methods.
- Experiments for continuous control is only conducted on Halfcheetah, a toy example that most RL baselines can easily solve.  Experiments on more complex tasks, i.e., humanoid-run, hopper-hop, acrobot-swingup, fish-swim, from dm_control would be more persuasive.

**Questions:**

- Since exploration is the natural setting for  λR, why there is no comparisons to different exploration baselines, i.e., RND, state entropy, pseudo-count ...?
- I am curious of how will the λR perform in more complex pixel-based tasks.

**Limitations:**

- The application of the proposed method seems to be narrow, and there is a lack of comparisons to SOTA exploration baselines.

---

> ### Author Rebuttal · Authors · 2023-08-09
>
>
> Thank you very much for your detailed review and feedback! We are glad that you appreciated the problem setting, found the writing to be clear, and valued our theoretical analysis. We aim to address your concerns below:
>
> - Typos: Thank you for pointing these out! We’ll correct them in the updated paper.
>
> - Baselines: We think there may be a misunderstanding of our experiments. While both the FR and the SR have been used as exploration bonuses, we don’t examine the $\lambda$R’s application to exploration in this paper, but rather its use for accurate policy evaluation, control, and composition in settings where agents experience diminishing marginal utility from repeated exposure to rewarding stimuli. We completely agree that had we used the $\lambda$R as an exploration bonus that these would be valuable baselines to compare against, but they aren’t as applicable in the experimental settings we do consider, which are primarily focused on establishing the usefulness of the $\lambda$R for supporting policy evaluation, control, and composition for problems with DMU. We completely agree that exploration is another interesting application area–we believe it is out of the scope of the current work, but would absolutely like to pursue it in the future.
>
> - Continuous Control Domains: Yes, this is a good point! We have added Hopper-v2 as an additional domain (Attachment Figures L3 and L4) and found that $\lambda$-SAC again matches SAC’s performance while being much less computationally demanding. Please see the general response above for a more detailed description.
>
> - Q1 (Exploration): See response above re: baselines. We are primarily concerned with problem settings involving diminishing marginal utility and show that in order to do RL using dynamic programming in such settings, the $\lambda$R is required. We didn’t offer comparisons to exploration baselines because we didn’t perform any experiments focused on exploration, but rather focused on establishing the $\lambda$R as useful for evaluation, control, and composition for settings with DMU.
>
> - Q2 (Further Pixel-Based Experiments): We certainly agree! We think the fact that the $\lambda$F was effective at facilitating pixel-based navigation is an encouraging first step, as successfully performing with 128 x 128 RGB observations is a good sign that the approach scales well. We hope to pursue further experimentation in the future!
>
>
> Thank you very much once again for your comments and questions! We hope our response has provided clarification and addressed your concerns. If that’s the case, we would greatly appreciate it if you would consider raising your score. If not, we would be more than happy to continue discussion!

---

> > ### Comment · Reviewer_2Ebj · 2023-08-18
> >
> > Thanks for the response. I have updated the score.

---

> > > ### Author Response · Authors · 2023-08-18
> > > **Thank you!**
> > >
> > > Thank you for your feedback and time! Please let us know if any other questions arise.

---

### Official Review · Reviewer_jk48 · 2023-07-06

**Soundness:** 4 excellent
**Presentation:** 4 excellent
**Contribution:** 3 good
**Rating:** 7
**Confidence:** 3

**Summary:**

The paper considers the case of non-Markov rewards, where rewards at states decay over time (motivated by the diminishing marginal utility phenomenon). While successor representations (SR) have been used in standard MDPs, the decaying of rewards means that they cannot be used here, and so the paper develops a "successor-like" representation that can handle this case, which generalises SR and similar other representations. Theoretical results ensure that these representations can be learned using a Bellman backup (as in the case of SR), which can be used to evaluate and improve policies. Further extensions to the function approximation case (where features are required) are also demonstrated, and bounds related to composition (similar to the successor features bound) are presented. Results in tabular and pixel-based domains showcase how the representation can be used with model-free RL algorithms to solve this class of tasks, where taking into account the decaying rewards (instead of assuming rewards are stationary) is shown to be beneficial.


**Strengths:**

The paper considers an interesting extension to the typical MDPs considered in reinforcement learning. The motivation regarding natural behaviours is an interesting one, but I think on a more fundamental level, any approach that attacks decision problems in RL that aren't Markovian in some way is of interest to the field.

This paper packs a lot into it - it develops many different ideas, ranging from the $\lambda$ representation in the tabular case to the function approximation "feature" case. It also shows how these can be used to evaluate and learn policies, compose existing value functions, and be applied in a deep RL context.

Aside from the extensive theory presented, the paper also does a good job of relating the approach to existing work in successor representations and demonstrating under what conditions that approach here simplifies to existing work. This helps to unify these otherwise disparate methods.

**Weaknesses:**

As presented, the experiments demonstrate that the method can be applied in both tabular and function approximation settings, that using the right value for $\lambda$ is necessary to achieve the best performance, and that composition can be achieved just as in the SR/SF case. While these experiments confirm the utility of the representation and go hand in hand with the theory, it would have been nice to see the performance of the approach compared to different baselines. This could answer questions like how it compares to SR/FR/SM under a variety of conditions: when $\lambda < 1$ or when $\lambda =1$ or when states are only ever visited once (because it's continuous/very high dimensional).

The pixel domain was helpful to showcase that the approach can be applied in high-dimensional domains like this, but it would have been nice to see a more "compositional" domain as has been demonstrated in previous work to investigate composition (e.g. the domains where the agent collects objects of different colours/shapes with different priorities) [1,2,3].

I realise space is an issue, but it would have been helpful to see a very brief discussion on related work that focuses on non-Markov rewards (outside of the context of successor representations). e.g. [4]

Minor:

Some of the drawbacks of composition with SF are also inherited by this approach. For instance, the bound in Theorem 5.1 is pretty loose (see [2]), but I understand the point is to show it's the same bound with an extra term that is positive only when the $\lambda$ estimates are wrong, so it's not very important

While I found the motivation for considering these forms of non-Markovian rewards interesting, it may be the case that many RL practitioners who are not interested in modelling animal/real-world behaviour will not adopt this since for many practical applications (or popular benchmarks like Atari or MuJoCo), such a formulation is not natural.

Some of the contractions used made it hard to parse sentences (e.g. L74). I'd recommend replacing these with the full words.

[1] Barreto, André, et al. "Fast reinforcement learning with generalised policy updates." Proceedings of the National Academy of Sciences 117.48 (2020): 30079-30087.

[2] Nangue Tasse, Geraud, et al. "Generalisation in lifelong reinforcement learning through logical composition." International Conference on Learning Representations. 2022.

[3] Alver, Safa, and Doina Precup. "Constructing a Good Behavior Basis for Transfer using Generalised Policy Updates." International Conference on Learning Representations. 2022.

[4] Gaon, Maor, and Ronen Brafman. "Reinforcement learning with non-markovian rewards." Proceedings of the AAAI conference on artificial intelligence. Vol. 34. No. 04. 2020.

**Questions:**

On line 291, it states that there is not yet a measure-theoretic version of the representation. Having said that, while there may not be strict theoretical justification, would there be any issue in practice with applying this approach to continuous state/action domains?

I'm slightly confused about the composition in the Four Rooms domain. There are four base policies but three goals. Is each policy to reach the centre of each room (and in three of the rooms, a goal exists)?

**Limitations:**

The limitations are well described in Section 7, including many that I would not have even considered.

---

> ### Author Rebuttal · Authors · 2023-08-09
>
> Thank you very much for your detailed feedback, and we’re glad that you liked the paper! We hope to address your concerns below:
>
> - Baselines: We completely agree, and identifying appropriate baselines was a challenge for us, simply because (as far as we know) DMU has not been addressed in an RL context before. Experiments where we set $\lambda = 1$ are meant to compare our approach to standard RL, but if you have any specific suggestions for additional baselines, we’d very much appreciate any pointers!
>
> - Compositional Domains: Thank you for pointing this out! We agree, one direction that we’d like to push in the future is to explore both more compositional tasks (e.g., environments where the agent has to pick up a key and proceed to a locked door) as well as novel ways to use the $\lambda$R to compose policies (e.g., the FR can be used to perform a form of shortest path planning over a set of base policies [1]).
>
> - Discussion of Non-Markov Rewards: Yes, we certainly agree–thank you for pointing this out! We will add a section in the Appendix with a more detailed discussion.
>
> - Looseness of the Bound: Yes, indeed it is a rather loose bound, but as you note, our hope was to draw the connection with the previously established bound for GPI :). We have slightly tightened the bound recently, replacing the $L_1(r)$ term with $L_\infty(r)$.
>
> - Interest to the Broader Community: We believe this is an important point, and one that we’ve given careful consideration. It is certainly the case that the most natural area of application for DMU is in studying natural behavior. In that lens, we hope this paper establishes a formal framework for DMU within RL that can be applied in the future more extensively within this context. However, we also believe that addressing non-stationarity in environmental rewards is also an important topic for the design of artificial agents. If agents are designed with the implicit assumption that rewards are ever-present, it could lead to value overestimation and suboptimal behavior. While this theme is present in all of our experiments, we’d like to highlight $\lambda$-SAC (Appendix H and Attachment Figures L3 and L4) as an interesting example case.
>
> - Contractions: Thank you! We will update the paper accordingly.
>
> - Q1 (Measure/Set-Theoretic Formulation in Practice): Good question! The only continuous-domain experiments that we performed were using $lambda$-features (Fig. 5.4), but we do not anticipate any issues with using the $\lambda$-operator in practice, even though it isn’t entirely theoretically justified. Continuous-domain experiments with the $\lambda$O are an exciting direction for future work that we intend to look at, especially because we could adjust FR planning (as mentioned above and detailed in [1]) to be compatible with the first-occupancy operator (i.e. $\lambda$O with $\lambda=0$) to construct a continuous-domain planning scheme. We can mention this in the paper.
>
> - Q2 (FourRooms): Yes, that’s correct! We’ll make it more clear in the text.
>
>
> [1] https://openreview.net/forum?id=JBAZe2yN6Ub

---

> > ### Comment · Reviewer_jk48 · 2023-08-12
> >
> > Thank you for the response. While there are still some fair questions about the motivation for the setting considered, and (as others have pointed out) many ways that the paper can be extended to incorporate other kinds of DMU-type problems, I think the paper packs a lot into it as it currently stands and am happy to stand by my original rating.

---

> > > ### Author Response · Authors · 2023-08-16
> > > **Thank you!**
> > >
> > > We appreciate your feedback, and thank you for the time spent reviewing the paper! Please let us know if any additional questions arise.

---

### Official Review · Reviewer_7pTK · 2023-07-12

**Soundness:** 3 good
**Presentation:** 3 good
**Contribution:** 3 good
**Rating:** 6
**Confidence:** 4

**Summary:**

The authors explore diminishing marginal utility in the context of reinforcement learning. Specifically, they study reinforcement learning when reward obtained at a state diminishes with each visit to that state (following a particular mathematical form). The authors show that, under such diminishing rewards, agent policies can be evaluated by using a novel state representation introduced in the paper. This representation generalises earlier state representations in the literature, such as the successor representation.

**Strengths:**

Originally: The concept of diminishing rewards is well known and explored in other fields. The paper presents new results within the context of reinforcement learning.

Quality: The authors provide a thoughtful and useful analysis under the assumptions they make. This includes the discussion of how the proposed representation relates to existing state representations in RL. The recursion they have identified is interesting and potentially useful.

The paper is well structured. The writing is clear.

**Weaknesses:**

The problem is not strongly motivated: why is studying diminishing marginal utility within the context of RL, with the specific assumptions made in this paper, important? Reasoning that goes beyond "people have diminishing marginal utility" would be useful. Why and how is this concept, in the form studied in the paper, useful for reinforcement learning agents?

The writing should clearly distinguish between two cases: (1) an objective quantity in the environment (e.g., available food) gets smaller, (2) the subjective value attached by the agent to an outside quantity gets smaller (e.g., the second ice cream cone does not taste as good).

In Section 6, the authors write that naturalistic environments often exhibit diminishing reward and give the example of foraging with diminishing food patches. I see a discrepancy between this example and the problem addressed in the rest of the paper. In the foraging example, the available food decreases because other agents consume it, and the amount of reduction should generally be a function of the time that passed between visits to the site. In contrast, the core assumption in the paper is that the reduction in reward in a given state is a function of how many times the agent itself visited that particular state, regardless of how much time passed since its last visit to that state.

Additional comments: Forward-backward representation should be explained in Section 2 (Preliminaries).

**Questions:**

I do not have any particular questions but, as I noted above, I do not find the paper particularly well-motivated in its present form. Why and how is diminishing marginal utility, in the form studied in the paper, useful for reinforcement learning agents?

I have read the rebuttal and thank the authors for their additional thoughts on the subject.

**Limitations:**

Yes, the limitations have been discussed.

---

> ### Author Rebuttal · Authors · 2023-08-09
>
> Thank you very much for your detailed review and careful consideration of our paper! We are glad that you appreciated the originality of the work, found it to be of high quality, well-structured, and clearly written. We aim to address your concerns below:
>
> - Motivation: We believe that there are two central questions that can be asked regarding the usefulness of DMU and RL. (1) Does incorporating DMU into standard RL paradigms/agents improve their performance on problems of interest in machine learning? We believe the following three areas are of particular importance in answering this question: (i) faster convergence for policy evaluation with lambda < 1 (Proposition 4.1 and Attachment Figure L1), reducing value function overestimation (e.g., $\lambda$-SAC, described in Appendix H and with additional results presented in Attachment Figures L3 and L4), (iii) greater understanding of non-stationary rewards in RL, and (iv) the set-theoretic formulation of the FR (lambda = 0)--suitable for continuous spaces–is novel. The FR facilitates navigation along shortest paths between goal states, which is of interest in robotics and other areas of machine learning. 2) Does modeling DMU within an RL framework provide new avenues of study for understanding natural behavior? This is the question for which we are most hopeful to provide an affirmative answer. The SR and FR are already of interest to neuroscientists [1,2,3] for the connections between the ways they encode spatial information and hippocampal activity. By generalizing these representations (and the problem settings for which they are suited) to DMU, we believe that the $\lambda$R can guide both further study in this area and offer new connections to other phenomena associated with DMU (e.g., foraging). Previous work on the SR and FR has already been grounded within the formalism of RL. When neuroscientists use these representations, then, they have clarity on what it is they can formally claim regarding behavior. We also believe these results may be of interest to economists interested in DMU. Our primary goal in this area is to provide the same type of formally sound framework for DMU as has been established for other problem areas, so that the $\lambda$R can act as a solid foundation on which to build. We view the experimental results in Section 6 as a proof of concept for the $\lambda$R as a useful tool in this setting. We will emphasize these motivations for the work in the paper.
>
> - The Nature of Diminishment: This is indeed a very good point, and one which we should emphasize and explore in more depth in the paper! It is especially interesting because in natural foraging, both the objective depletion of resources in the environment and subjective diminishment of perceived utility play a role in behavior (e.g., an animal may leave a berry patch before all berries have been consumed if another resource’s appeal supersedes the consumption of more fruit). One especially important point that you bring up is that of utility replenishment–i.e., that utility should be a function of the amount of time since the agent has last visited a particular state rather than just how many times it’s visited that state overall. As we note in our discussion of limitations in Section 7, the current framework we present models this replenishment process as being linked to the episodic nature of the environment, but in an ideal case, replenishment would take place gradually in a continuing environment. We explore this case in a fair amount of detail in Appendix I, providing some possible representations in line with such a framework as well as preliminary simulations. We agree this is the natural next step for studying DMU and foraging within RL!
>
> - FB Representation Background: Yes, that is a good point–we will make the necessary adjustments.
>
> Thank you very much once again for the helpful feedback! We hope we have adequately addressed your concerns. If not, we’re more than happy to continue the discussion!
>
> [1] https://www.nature.com/articles/nn.4650
>
> [2] https://www.jneurosci.org/content/38/33/7193
>
> [3] https://www.cell.com/neuron/pdf/S0896-6273(23)00230-1.pdf

---

### Author Rebuttal · Authors · 2023-08-09


We’d like to sincerely thank all reviewers for their time and for their helpful feedback and suggestions for the paper. We believe that incorporating this feedback will make the paper stronger, and we look forward to a constructive discussion! We are glad that reviewers consistently appreciated the originality of the paper, the theoretical analysis provided, the consideration of both tabular and deep RL instantiations of our approach, and found the writing to be clear. Overall, we view this as exploratory work—DMU is a prominent factor in human and animal decision-making. RL is likely the most dominant approach to modeling sequential decision-making in machine learning and a common approach for doing so in neuroscience and other fields, yet until now (as far as we are aware) has not contained a framework that accounts for (and formally characterizes) DMU. The goal of this paper is to provide a basic approach for closing this gap.

We have also attached a pdf file with several additional experimental results:

- In Figure L1, we demonstrate the accelerated convergence of dynamic programming—in this case, policy evaluation—for lower values of $\lambda$ both with and without stochastic transition dynamics in the environment (in this case, FourRooms). This result augments the one in Figure E3.

- In Figure L2, we show that the usefulness of the $\lambda$R for supporting policy composition is maintained with stochastic transition dynamics.

- In Figure L3, we augment the results in Figure H1 in the following ways: (1) we add five additional random seeds to our HalfCheetah-v2 results, and (2) we repeat our experiment on Hopper-v2, a more challenging environment (also for eight seeds total). We found that, consistent with our initial results, $\lambda$-SAC is able to match the performance of SAC while only using a single critic (rather than two). It does this by using an online bandit approach to select values of $\lambda$ used for training the critic that result in higher reward (see Appendix H for more details). In the rightmost panel, we can see how the choice of $\lambda$ changes over the course of learning for each environment. Interestingly, in [1], it was found that strong performance in HalfCheetah-v2 is associated with optimistic/high value estimates, while strong performance in Hopper-v2 is associated with pessimistic/lower value estimates (i.e., it benefits more from the pessimistic evaluation performed when SAC takes the minimum across its two critics). Consistent with this finding, we observed that $\lambda$-SAC learns to select high values of $\lambda$ on HalfCheetah-v2, and lower values on Hopper-v2. We believe this is a promising example of a possible area of benefit that the $\lambda$R can bring even to standard RL problem settings where the reward is stationary.

- In Figure L4, we plot the average final performance of SAC and $\lambda$-SAC versus the FLOPS used by each agent (calculated using the `flopth` Python package) as a means of emphasizing that $\lambda$-SAC is able to match SAC’s performance while substantially saving on computational cost (~32% for each task).

We aim to address specific concerns in the responses below. In these responses, when we refer to the additional figures described above, we will name them as “Attachment Figures.” Thank you all once again!

[1] https://openreview.net/forum?id=a4WgjcLeZIn

---

### Decision · Program_Chairs · 2023-09-21

**Decision:**

Accept (poster)

**Comment:**

Paper looks at diminishing marginal utility and how this is factored into an MDP setting, which has been done before (as the authors state) but has novelty here with in terms of RL applications by generalizing the successor representation. Some of the experimental setups arguably could have used more baselines. The pixel-based experiments showcase the authors' method quite well. Reviewer questions mostly answered.